# FREQUENCY-AWARE SGD FOR EFFICIENT EMBEDDING LEARNING WITH PROVABLE BENEFITS

**Yan Li** [*]
ISyE, Georgia Tech
yli939@gatech.edu

**Dhruv Choudhary**
Meta
choudharydhruv@fb.com

**Xiaohan Wei**
Meta
ubimeteor@fb.com

**Baichuan Yuan**
Meta
bcyuan@fb.com

**Bhargav Bhushanam**
Meta
bbhushanam@fb.com

**Tuo Zhao**
ISyE, Georgia Tech
tourzhao@gatech.edu

**Guanghui Lan**
ISyE, Georgia Tech
george.lan@isye.gatech.edu

## ABSTRACT

Embedding learning has found widespread applications in recommendation systems and natural language modeling, among other domains. To learn quality embeddings efficiently, adaptive learning rate algorithms have demonstrated superior empirical performance over SGD, largely accredited to their token-dependent learning rate. However, the underlying mechanism for the efficiency of token-dependent learning rate remains underexplored. We show that incorporating frequency information of tokens in the embedding learning problems leads to provably efficient algorithms, and demonstrate that common adaptive algorithms implicitly exploit the frequency information to a large extent. Specifically, we propose (Counter-based) Frequency-aware Stochastic Gradient Descent, which applies a frequency-dependent learning rate for each token, and exhibits provable speed-up compared to SGD when the token distribution is imbalanced. Empirically, we show the proposed algorithms are able to improve or match adaptive algorithms on benchmark recommendation tasks and a large-scale industrial recommendation system, closing the performance gap between SGD and adaptive algorithms, while using significantly lower memory. Our results are the first to show token-dependent learning rate provably improves convergence for non-convex embedding learning problems.

## 1 INTRODUCTION

Embedding learning describes a problem of learning dense real-valued vector representation for categorical data, often referred to as token (Pennington et al., 2014; Mikolov et al., 2013a;b). Good quality embeddings can capture rich semantic information of tokens, and thus serve as the cornerstone for downstream applications (Santos et al., 2020). Due to their significant impact on model performance and large memory footprint (21.8% of total parameters for BERT (Devlin et al., 2018), 95% for industrial recommenders in Section 4), how to learn quality embedding vectors efficiently forms an important problem in applications, including recommendation systems and natural language processing.

Empirically, adaptive algorithms (Duchi et al., 2011; Kingma & Ba, 2014; Reddi et al., 2019) have witnessed significant successes, yielding state of the art performance in both industrial-scale recommendation systems and natural language model (Guo et al., 2017; Zhou et al., 2018b; Devlin et al., 2018; Liu et al., 2019). Stochastic gradient descent (SGD), on the other hand, has struggled to keep up, often yielding much slower convergence and low quality models (Liu et al., 2020; Zhang et al., 2019) (see also Figure 2). The sharp contrast on the efficiency of adaptive algorithms and SGD is particularly distinctive, as SGD is the typical choice of optimization algorithms in the other domains of machine learning, such as vision/image related tasks (He et al., 2016; Goyal et al., 2017).

---

[*]Work done during an internship at Meta.

The common belief behind the empirical edge of adaptive learning rate algorithms over SGD is that the former ones exploit sparsity of high dimensional feature. Specifically, a feature in a typical embedding learning problem comes in the form of one/multi-hot encoding of tokens (e.g. wordpiece in NLP and user/item in recommendation systems), which leads to a sparse stochastic gradient that has only non-zero values for tokens within the mini-batch. In addition, token distributions of real world data are often highly imbalanced and satisfy the power-law property (Piantadosi, 2014; Celma, 2010; Clauset et al., 2009), and infrequent tokens are widely believed to be more informative to model learning. Thus adaptive algorithms can pick up information from the infrequent tokens more efficiently, as they can schedule a higher learning rate for the infrequent tokens (Duchi et al., 2011).

Despite the appealing intuition, there is a significant theory-practice gap on the empirical superiority of adaptive learning rate algorithms over SGD, and no developed theories can explicitly justify the previous intuition. Better dimensional dependence of adaptive algorithms has only been shown in the convex setting (Duchi et al., 2011), which hardly generalizes to even the simplest practical models in embedding learning problems (e.g., Factorization Machine, Rendle (2010)), whose loss landscape is non-convex. For non-convex settings, most theoretical efforts have been devoted to analyzing adaptive learning rate algorithms for general non-convex objectives, which yield subpar convergence rate compared to standard SGD (Ward et al., 2018; Défossez et al., 2020; Chen et al., 2018; Zhou et al., 2018a). In fact, the standard SGD has been recently shown to be minimax optimal for non-convex problems (Drori & Shamir, 2020; Arjevani et al., 2019), and thus not improvable in general. Moreover, since adaptive algorithms are only implicitly exploiting frequency information, and if the intuition indeed holds true, one might naturally wonder whether we can instead develop an adaptive learning rate schedule that explicitly depends on frequency information. Motivated by our previous discussions, we raise and aim to address the following questions

> **Questions**
>
> *Can we design a frequency-dependent adaptive learning rate schedule? Can we show provable benefits over SGD?*

**Our contributions.** We answer the previous question by showing that token frequency information can be leveraged to design provably efficient algorithms for embedding learning. Specifically,

- We propose **F**requency-**a**ware **S**tochastic **G**radient **D**escent (FA-SGD), a simple modification to standard SGD, which applies a token-dependent learning rate that inversely proportional to the frequency of the token. We also propose a variant, named **C**ounter-based **F**requency-aware **S**tochastic **G**radient **D**escent (CF-SGD), which is able to estimate frequency in an online fashion, much similar to Adagrad (Duchi et al., 2011) and Adam (Kingma & Ba, 2014).
- Theoretically, we show that both FA-SGD and CF-SGD outperform standard SGD for embedding learning problems. Specifically, they are able to significantly improve convergence for learning infrequent tokens, while maintaining convergence speed for frequent tokens. To the best of our knowledge, our proposed algorithms are the first to show provable speed-up over standard SGD for non-convex embedding learning problems. This is in sharp contrast with other popular adaptive learning rate algorithms, whose empirical performance can not be explained by existing theories.
- Empirically, we conduct extensive experiments on benchmark datasets and a large-scale industrial recommendation system. We show that FA/CF-SGD is able to significantly improve over SGD, and improves/matches popular adaptive learning rate algorithms. We also observe the second-order moment maintained by Adagrad and Adam highly correlates with the frequency information, demonstrating intimate connections between adaptive algorithms and the proposed FA/CF-SGD.

## 1.1 RELATED LITERATURE

**Adaptive algorithms for non-convex problems.** There has been a fruitful line of research on analyzing the convergence of adaptive learning rate algorithms in non-convex setting. These results aim to match the convergence rate of standard SGD given by $\mathcal{O}(1/\sqrt{T})$ (Ghadimi & Lan, 2013), however often with additional factor of $\log T$ (Ward et al., 2018; Défossez et al., 2020; Chen et al., 2018; Reddi et al., 2018), or with worse dimension dependence (Zhou et al., 2018a) for smooth problem (assumed by almost all prior works). Moreover, all existing works aim to analyze the convergence for general non-convex problems, ignoring unique data features in embedding learning problems, where adaptive algorithms are most successful. We explicitly take account into the spar-

---

**Algorithm 1** Frequency-aware Stochastic Gradient Descent

---

**Input:** Total iteration number $T$, token frequency $\{p_k\}_{k \in X}$, and learning rate schedule $\{\eta_k^t\}_{k \in X, t \in [T]}$ specified by (7).
**Initialize:** $\Theta^0 \in \mathbb{R}^{N \times d}$, sample $\tau \sim \text{Unif}([T])$,
**for** $t = 0, \ldots \tau$ **do**
    (1) Sample $(i_t, j_t) \sim \mathcal{D}$, calculate $g_{i_t}^t = \nabla_{\theta_{i_t}} \ell(\theta_{i_t}, \theta_{j_t}; y_{i_t, j_t})$, $g_{j_t}^t = \nabla_{\theta_{j_t}} \ell(\theta_{i_t}, \theta_{j_t}; y_{i_t, j_t})$
    (2) Update parameters
$$\theta_{i_t}^{t+1} = \theta_{i_t}^t - \eta_{i_t}^t g_{i_t}^t, \;\; \theta_i^{t+1} = \theta_i^t, \;\; \forall i \in U, i \neq i_t$$
$$\theta_{j_t}^{t+1} = \theta_{j_t}^t - \eta_{j_t}^t g_{j_t}^t, \;\; \theta_j^{t+1} = \theta_j^t, \;\; \forall j \in V, j \neq j_t$$
**end for**
**Output:** $\Theta^\tau$

---

sity of stochastic gradient, and token distribution imbalancedness into the design and analysis of our proposed algorithms, which are the keys to better convergence properties.

**Adaptive algorithms and SGD.** To the best of our knowledge, the study on understanding why adaptive learning rate algorithms outperform SGD is very limited. Zhang et al. (2019) argue that BERT pretraining (Devlin et al., 2018) has heavy-tailed noise, implying unbounded variance and possible non-convergence of SGD. Normalized gradient clipping method is proposed therein and converges for a family of heavy-tailed noise distributions. Our results focus on a different direction by showing that imbalanced token distribution is an important factor that can be leveraged to design more efficient algorithms for embedding learning problems. Our result also does not rely on the noise to be heavy-tailed for the convergence benefits of the proposed FA/CF-SGD to take effect.

**Notations:** For a vector/matrix, we use $\|\cdot\|$ to denotes its $\ell_2$-norm/Frobenius norm. We use $\|\cdot\|_2$ to denote the spectral norm of a matrix.

## 2 PROBLEM SETUP

We consider an embedding learning problem which aims to learn user and item embeddings through their interactions. We denote $U$ as the set of users, and $V$ as the set of items, and let $X = U \cup V$ denote the union, referred to as tokens throughout the rest of the paper. We assume $|X| = N$, i.e., the total number of user and item is $N$. For the ease of presentation, we always use letter $i$ to index user set $U$, letter $j$ to index item set $V$, and letter $k$ to index the union set $X$. The embedding learning problem can be abstracted into the following stochastic optimization problem:

$$\min_{\Theta \in \mathbb{R}^{N \times d}} f(\Theta) = \mathbb{E}_{(i,j) \sim \mathcal{D}} [\ell(\theta_i, \theta_j; y_{ij})] = \sum_{i \in U, j \in V} D(i,j) \ell(\theta_i, \theta_j; y_{ij}). \tag{1}$$

Here $(i, j)$ denotes the user-item pair sampled from the unknown interaction distribution $\mathcal{D}$, $\theta_i$, $\theta_j \in \mathbb{R}^d$ (the $i, j$-th row of $\Theta$) denotes their embedding vectors respectively, and the loss $\ell(\theta_i, \theta_j; y_{ij})$ denotes the prediction loss for their interaction $y_{ij} \in \{-1, +1\}$ (e.g., logistic loss). We further let

$$p_i = \sum_{j \in V} D(i,j), \;\; \forall i \in U; \;\; p_j = \sum_{i \in U} D(i,j), \;\; \forall j \in V, \tag{2}$$

denote the marginal distribution over $U$ and $V$.

*Remark* 2.1. Our analysis also allows treatment of additional network structure (with parameters denoted by $\mathcal{W}$) that takes nonlinear transformation of embedding vectors, e.g., $f(\Theta, \mathcal{W}) = \mathbb{E}_{(i,j) \sim \mathcal{D}} \ell(\theta_i, \theta_j, \mathcal{W}; y_{ij})$. We omit their explicit treatment for presentation simplicity. In addition, although we mainly discuss in the context of recommendation, our analysis and results only relies on sparsity of stochastic gradient and the imbalancedness of token distributions, which allow one to extend our results to other embedding learning problems (e.g., language model pretraining).

The full algorithmic descriptions of our proposed Frequency-aware Stochastic Gradient Descent (FA-SGD) algorithm are presented in Algorithm 1. Note that randomly outputting a historical iterate is commonly adopted in literature for showing convergence of stochastic gradient descent type algorithms for non-convex problems (Ghadimi & Lan, 2013). In practice, we can simply use the last iterate $\Theta^T$ as the output solution. In addition, Section 3.3 presents CF-SGD (Algorithm 2), which does not need the token distribution as the input and can estimate it in an online fashion.

At iteration $t$, FA-SGD samples $(i_t, j_t) \sim \mathcal{D}$, and obtain the sparse stochastic gradient $g^t$ defined in (3). Note that only the $i_t$-th and $j_t$-th row of $g_t$ are non-zero. One can readily verify that $\mathbb{E}_{(i_t, j_t) \sim \mathcal{D}} [g_t] = \nabla_\Theta f(\Theta^t)$. Going forward, we will denote $\nabla f_k^t$ as the $k$-th row of gradient $\nabla f(\Theta^t)$, and $g_k^t$ as the $k$-th row of stochastic gradient $g_t$. Note that we have

$$g_t = \begin{bmatrix} \mathbf{0}^\top \\ \nabla_{\theta_{i_t}} \ell(\theta_{i_t}, \theta_{j_t}; y_{i_t, j_t})^\top \\ \vdots \\ \nabla_{\theta_{j_t}} \ell(\theta_{i_t}, \theta_{j_t}; y_{i_t, j_t})^\top \\ \mathbf{0}^\top \end{bmatrix} \qquad (3)$$

$$\mathbb{E}_{j_t} \left[ g_{i_t}^t | i_t = i \right] = \nabla f_i^t / p_i, \quad \mathbb{E}_{i_t} \left[ g_{j_t}^t | j_t = j \right] = \nabla f_j^t / p_j. \qquad (4)$$

We further denote $\delta_k^t = \frac{1}{p_k} f_k^t - g_k^t$ for all $k \in X$. Then by definition $\mathbb{E} \left[ \delta_{i_t}^t | i_t = i \right] = 0$ and $\mathbb{E} \left[ \delta_{j_t}^t | j_t = j \right] = 0$ for all $i \in U, j \in V$. We pose the following assumptions on the its variance.

**Assumption 1** (Bounded conditional variance). *We assume that the variance of $\delta_{i_t}^t$ is bounded. That is, there exists $\{\sigma_k^2\}_{k \in X}$, such that*

$$\mathbb{E} \left[ \|\delta_{i_t}\|^2 | i_t = i \right] \le \sigma_i^2, \quad \mathbb{E} \left[ \|\delta_{j_t}\|^2 | j_t = j \right] \le \sigma_j^2, \quad \forall i \in U, j \in V. \qquad (5)$$

Assumption 1 allows us to provide a finer characterization on the variance of stochastic gradient compared to typical variance assumption in literature. To illustrate, recall that standard assumption in the stochastic optimization literature assumes $\mathrm{Var}(g^t) = \mathbb{E} \|\nabla_\Theta f^t - g^t\|^2 \le \sigma^2$ for some universal constant $\sigma > 0$. Consider an extreme setting, where we have exact gradient for the sampled user-item pair, i.e., $g_{i_t}^i = \frac{1}{p_{i_t}} \nabla f_{i_t}^t$ and $g_{j_t}^i = \frac{1}{p_{j_t}} \nabla f_{j_t}^t$, then we have $\sigma_k = 0$ for all $k \in X$. In contrast, the variance of $g^t$ is still non-zero. In general setting, we can bound the variance as shown in the following proposition. Note that the variance lower bound arises naturally form the extreme sparsity of the stochastic gradient.

**Proposition 2.1.** *Given Assumption 1, we have*

$$\sum_{k \in X} (1/p_k - 1) \left\| \nabla f_k^t \right\|^2 \le \mathrm{Var}(g^t) \le \sum_{k \in X} p_k \sigma_k^2 + \sum_{k \in X} (1/p_k - 1) \left\| \nabla f_k^t \right\|^2. \qquad (6)$$

**Assumption 2** (Smoothness of prediction loss). *We assume $\ell(u, v; y)$ is symmetric w.r.t. $u$ and $v$ for any $y \in \{-1, +1\}$, and there exists $L > 0$ such that $\left\| \nabla_{uu}^2 \ell(\cdot, \cdot; \cdot) \right\|_2 \le L, \left\| \nabla_{uv}^2 \ell(\cdot, \cdot; \cdot) \right\|_2 \le L$.*

The assumption on the symmetry of $\ell$ is readily satisfied by almost all neural network architecture. In essence, this assumption only requires that the parameterization of embedding vector is token agnostic. On the other hand, the spectral upper bound on the Hessian matrix is a standard assumption in optimization literature.

## 3 THEORETICAL RESULTS

We first present the convergence results of FA-SGD and standard SGD for embedding learning problem formulated in (1), and discuss the advantage that FA-SGD offers when the token distribution $\{p_k\}_{k \in X}$ is highly imbalanced. We further propose a variant, named CF-SGD, which can estimate frequency information in an online fashion and still provably enjoys the benefits of FA-SGD.

### 3.1 CONVERGENCE OF FA-SGD AND STANDARD SGD

**Theorem 3.1** (FA-SGD). *With Assumption 1 and 2, take learning rate policy to be*

$$\eta_k^t = \min \left\{ 1/(4L), \alpha/\sqrt{T p_k} \right\}, \qquad (7)$$

*where $T$ denotes the total number of iterations, and $\alpha = \sqrt{(f(\Theta^0) - f^*) / \left( L \sum_{l \in X} p_l \sigma_l^2 \right)}$, we have*

$$\mathbb{E} \|\nabla f_k^\tau\|^2 = \mathcal{O} \left( \frac{L(f(\Theta^0) - f^*)}{T} + \frac{\sqrt{p_k} \sqrt{\sum_{l \in X} p_l \sigma_l^2 (f(\Theta^0) - f^*) L}}{\sqrt{T}} \right), \quad \forall k \in X. \qquad (8)$$

*Remark* 3.1 (Connection with Stochastic Block Coordinate Descent). Our FA-SGD shares some similarities with Stochastic Block Coordinate Descent (SBCD) (Nesterov, 2012; Dang & Lan, 2015; Richtárik & Takáč, 2014) applied to problem (1), in the sense that each iteration we sample certain blocks of variables ($\theta_{i_t}, \theta_{j_t}$ in our case), and only update the sampled blocks by following its stochastic gradient. Different from SBCD, the stochastic gradient of the block variable $g_{i_t}^t$ in the FA-SGD is *biased*, as shown in (4). Note that with unbiased stochastic gradient, SBCD method typically converges slower than standard SGD by a factor that can be as large as number of blocks. As a concrete example, when the token distribution is uniform, SBCD converges slower than standard SGD by a factor of $|X|$, hence slower than FA-SGD by a factor of $|X|$ from Corollary 3.1 developed later.

Recall that from Proposition 2.1, the variance of stochastic gradient is heavily influenced by the population gradient $\nabla_\Theta f(\Theta)$, and can be huge whenever the population gradient is, presumably in the early phase of training. This relationship is also supported by empirical findings in Zhang et al. (2019) (Figure 2a), where the authors show that for BERT pretraining, the noise distribution in stochastic gradient $g^t$ is highly non-stationary, which has large variance in the beginning of the training and smaller variance at the end of training. Since existing analysis of SGD in literature assumes a constant variance bound for the stochastic gradient, our observation in Proposition 2.1 requires an alternative analysis of SGD for problem (1).

To obtain the convergence rate of standard SGD in the presence of iterate-dependent variance (6), our key insight is to tailor the convergence analysis to the sparsity of the stochastic gradient for problem (1). We show the convergence of standard SGD as the following.

**Theorem 3.2** (Standard SGD). *With Assumption 1 and 2, take learning rate policy to be* $\eta_k^t = \min\left\{\frac{1}{4L}, \frac{\alpha}{\sqrt{T}}\right\}$, *where $T$ denotes the total number of iterations, and* $\alpha = \sqrt{\frac{f(\Theta^0) - f^*}{L \sum_{l \in X} p_l^2 \sigma_l^2}}$, *we have*

$$\mathbb{E}\|\nabla f_k^\tau\|^2 = \mathcal{O}\left(\frac{L(f(\Theta^0) - f^*)}{T} + \frac{\sqrt{\sum_{l \in X} p_l^2 \sigma_l^2 (f(\Theta^0) - f^*)L}}{\sqrt{T}}\right), \quad \forall k \in X. \quad (9)$$

Note that both FA-SGD and standard SGD attain a rate of $\mathcal{O}(1/\sqrt{T})$. Compared to existing rates of standard SGD (Ghadimi & Lan, 2013), we do not require constant variance bound on stochastic gradient, as we have discussed above. Compared to existing rates of adaptive learning rate algorithms (Zhou et al., 2018a; Chen et al., 2018), both rates obtained here exhibits *dimension-free* property. We emphasize here that due to the dimension-free nature of the bounds for both SGD and FA-SGD, *we do not claim the proposed FA-SGD has better dependence on dimension, which is the main motivation of adaptive algorithms* (Duchi et al., 2011; Kingma & Ba, 2014; Reddi et al., 2019). Instead, the major difference on the convergence of FA-SGD (8) and that of standard SGD (9) is that the former one is *token-dependent*. Specifically, for FA-SGD, each token $k \in X$ has its own convergence characterization, while all the tokens have the same convergence characterization in the standard SGD. We first make a simple observation stating the equivalence of FA-SGD and standard SGD, when the token distribution $\{p_k\}_{k \in X}$ is uniform.

**Corollary 3.1** (Uniform Distribution). *Suppose the user distribution $\{p_i\}_{i \in U}$ and item distribution $\{p_j\}_{j \in V}$ is the uniform distribution. Then FA-SGD and standard SGD is equivalent to each other, in terms of both algorithmic execution and convergence rate.*

## 3.2 WHEN DOES FA-SGD OUTPERFORM STANDARD SGD?

We show FA-SGD shines when the token distribution $\{p_k\}_{k \in X}$, defined in (2), is highly imbalanced. Before we present detailed discussions, we make an important remark that highly imbalanced token distributions are ubiquitous in social systems, presented in the form *power-law*. Examples of such distributions include the degree of individuals in the social network (Muchnik et al., 2013); the frequency of words in natural language (Zipf, 2016); citations for academic papers (Brzezinski, 2015); number of links on the internet (Albert et al., 1999). For more discussions on power-law distributions in social and natural systems, we refer readers to Kumamoto & Kamihigashi (2018).

In Figure 1c, 1d we plot the user and item counting distribution of Movielens-1M dataset. One could clearly see that the user and item distributions are highly imbalanced, with a small percentages of users/items taking up the majority of rating records. We defer details on the skewness of token distributions for Criteo dataset to Appendix C.

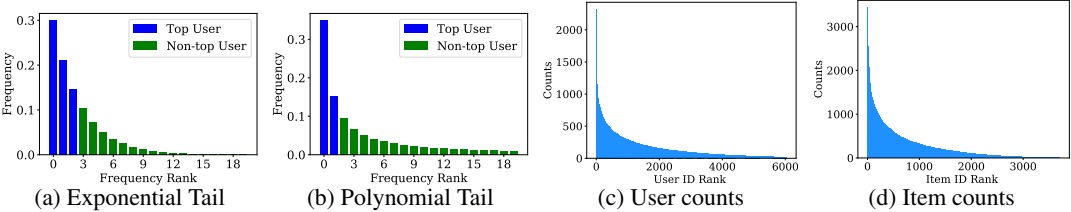

Figure 1: Token distribution with an exponential and polynomial tail, and the user/item counting distributions for Movielens-1M dataset.

To illustrate the comparative advantage of FA-SGD when the token distribution $\{p_k\}_{k \in X}$ is highly skewed. We consider two classes of distribution families with different tail properties, one with exponential tail, and one with polynomial tail.

**Corollary 3.2** (Exponential Tail). *Let* $U = \{i_n\}_{n=1}^{|U|}$, $V = \{j_m\}_{m=1}^{|V|}$, *where* $i_n$ *denote the user with* $n$*-th largest frequency, and* $j_m$ *denote the item with the* $m$*-th largest frequency. Suppose*

$$p_{i_n} \propto \exp(-\tau n), \quad p_{j_m} \propto \exp(-\tau m), \quad \forall n \in [[U]], m \in [[V]] \tag{10}$$

*for some* $\tau > 0$. *Define* $U_T$ *as the set of users whose frequencies are within* $e$*-factor from the highest frequency:* $U_T = \{i_n : n \leq \frac{1}{\tau}\}$, *and* $V_T$ *similarly as* $V_T = \{j_m : m \leq \frac{1}{\tau}\}$. *We refer to* $U_T$ *as the top users, and* $V_T$ *as the top items.*

*Then given* $|U|, |V| \geq \frac{1}{\tau}$, *the proposed FA-SGD, compared to standard SGD:*

*(1) Obtains the same rate of convergence, for the top users* $U_T$ *and top items* $V_T$;

*(2)* $\mathbb{E} \left\| \nabla f_{i_n}^\tau \right\|^2$ *can converge faster by a factor of* $\Omega \{\exp (\tau(n - |U_T|))\}$ *for* $i_n \in U \setminus U_T$ ;

*(3)* $\mathbb{E} \left\| \nabla f_{j_m}^\tau \right\|^2$ *can converge faster by a factor of* $\Omega \{\exp (\tau(m - |V_T|))\}$ *for* $j_m \in V \setminus V_T$.

We remark that $|U|, |V| \geq \frac{1}{\tau}$ is a very mild condition, as it only requires that the most infrequent user/item should have its frequency smaller than the most frequent user/item by at least a factor of $e$. i.e., the non-top user/item set $U \setminus U_T$, $V \setminus V_T$ is nonempty, This is readily satisfied by the token distributions in recommendation systems and natural language modeling (Celma, 2010; Zipf, 2016), where the lowest frequency is at least orders of magnitude smaller than the highest frequency. The factor of $e$ in defining $U_T, V_T$ can also be readily replaced by any constant larger than 1.

From Corollary 3.2, we can see that FA-SGD improves significantly over standard SGD for user/item distribution with exponential tail. Specifically, FA-SGD achieves the same convergence rate of top users/items compared to SGD, meanwhile it significantly improves the convergence of the non-top users/items. Moreover, the strength of such an *improvement increases exponentially as we move towards the tail users/items.*

**Corollary 3.3** (Polynomial Tail). *Let* $U = \{i_n\}_{n=1}^{|U|}$, $V = \{j_m\}_{m=1}^{|V|}$, *where* $i_n$ *denote the user with* $n$*-th largest frequency, and* $j_m$ *denote the item with the* $m$*-th largest frequency. Suppose*

$$p_{i_n} \propto n^{-\nu}, \quad p_{j_m} \propto m^{-\nu}, \quad \forall n \in [[U]], m \in [[V]] \tag{11}$$

*for some* $\nu \geq 2$. *Define* $U_T$ *as the set of users whose frequencies are within* 2*-factor from the highest frequency:* $U_T = \{i_n : n^{-\nu} \geq 1/16\}$, *and* $V_T$ *similarly as* $V_T = \{j_m : m^{-\nu} \geq 1/16\}$. *We refer to* $U_T$ *as the top users, and* $V_T$ *as the top items.*

*Then given* $|U|, |V| \geq 16^{1/\nu}$, *the FA-SGD, compared to standard SGD:*

*(1) Obtains the same rate of convergence, for the top users* $U_T$ *and top items* $V_T$;

*(2)* $\mathbb{E} \left\| \nabla f_{i_n}^\tau \right\|^2$ *can converge faster by a factor of* $\Omega \left\{ \left( \frac{n}{|U_T|} \right)^\nu \right\}$ *for each* $i_n \in U \setminus U_T$;

*(3)* $\mathbb{E} \left\| \nabla f_{j_m}^\tau \right\|^2$ *can converge faster by a factor of* $\Omega \left\{ \left( \frac{m}{|V_T|} \right)^\nu \right\}$ *for each* $j_m \in V \setminus V_T$.

We remark that polynomial tail (37) is also the prototypical example of the power law distribution class for modeling social behaviors (Kumamoto & Kamihigashi, 2018). The constant 2 in the condition $\nu \geq 2$ can be replaced by any constant strictly larger than 1, with slight changes to the constant factor in the statements of the corollary.

From Corollary 3.3, we can see that FA-SGD improves significantly over standard SGD for user/item distribution with polynomial tail. Specifically, FA-SGD achieves the same convergence rate of top

---

**Algorithm 2** Counter-based Frequency-aware Stochastic Gradient Descent

---

**Input:** Total iteration number $T$.
**Initialize:** $\Theta^0 \in \mathbb{R}^{N \times d}$, counter sample $\tau \sim \mathrm{Unif}(\{T/2, \dots, T\})$.
**for** $t = 0, \dots \tau$ **do**
    (1) Sample $(i_t, j_t) \sim \mathcal{D}$, calculate $g_{i_t}^t = \nabla_{\theta_{i_t}} \ell(\theta_{i_t}, \theta_{j_t}; y_{i_t, j_t})$, $g_{j_t}^t = \nabla_{\theta_{j_t}} \ell(\theta_{i_t}, \theta_{j_t}; y_{i_t, j_t})$
    (2) Compute counter-based learning rate $\widehat{\eta}_{i_t}^t(c_{i_t}^t)$, $\widehat{\eta}_{j_t}^t(c_{j_t}^t)$ specified by (12)
    (3) Update parameters

$$\theta_{i_t}^{t+1} = \theta_{i_t}^t - \widehat{\eta}_{i_t}^t g_{i_t}^t, \ \ \theta_i^{t+1} = \theta_i^t, \ \ \forall i \in U, i \neq i_t$$
$$\theta_{j_t}^{t+1} = \theta_{j_t}^t - \widehat{\eta}_{j_t}^t g_{j_t}^t, \ \ \theta_j^{t+1} = \theta_j^t, \ \ \forall j \in V, j \neq j_t$$

    (4) Update counters

$$c_{i_t}^{t+1} = c_{i_t}^t + 1, \ \ c_i^{t+1} = c_i^t, \ \ \forall i \in U, i \neq i_t$$
$$c_{j_t}^{t+1} = c_{j_t}^t + 1, \ \ c_j^{t+1} = c_j^t, \ \ \forall j \in V, j \neq j_t$$

**end for**
**Output:** $\Theta^\tau$

---

users/items compared to SGD, meanwhile it significantly improves the convergence of the non-top users/items. Moreover, the strength of such an *improvement increases in polynomial order as we move towards the tail users/items.*

### 3.3 ONLINE ESTIMATION OF FREQUENCY INFORMATION

In certain application scenarios, the token distribution $\{p_k\}_{k \in X}$ can be unknown in advance of learning. To apply FA-SGD, one needs to employ a preprocessing step in order to estimate the token distribution to a high accuracy, and then run the algorithm with estimated token distribution. Such a preprocessing step often requires additional human efforts and data. To remove such an undesirable preprocessing step, below we present an online variant of FA-SGD, which uses the counter of tokens collected during training to estimate the token distribution dynamically. We show that the proposed **C**ounter-based **F**requency-aware **S**tochastic **G**radient **D**escent (CF-SGD) is able to retain the benefits of FA-SGD despite unknown token distribution.

**Theorem 3.3** (Counter-based FA-SGD). *In addition to Assumption 1 and 2, suppose $\|\nabla f(\cdot)\| \leq G$. Take counter-based learning rate policy in Algorithm 2 to be*

$$\widehat{\eta}_k^t(c_k^t) = \min\left\{ 1/(4L), 1/\sqrt{T \widehat{p}_k^t} \right\}, \ \ \widehat{p}_k^t = c_k^t/t, \ \ \forall k \in X, t \in [T], \tag{12}$$

*where $T$ denotes the total number of iterations, $\alpha = \sqrt{M_f / \left(L \sum_{l \in X} p_l \sigma_l^2\right)}$ and $M_f = f(\Theta^0) - f^* + \sum_{k \in X} p_k \sigma_k^2 / L$, we have*

$$\mathbb{E} \|\nabla f_k^\tau\|^2 = \mathcal{O}\left( \frac{L M_f}{T} + \frac{\sqrt{p_k}\sqrt{\sum_{l \in X} p_l \sigma_l^2 L(f(\Theta^0) - f^*)}}{\sqrt{T}} + \frac{\sqrt{p_k}\left(\sum_{l \in X} p_l \sigma_l^2\right)}{\sqrt{T}} \right), \forall k \in X. \tag{13}$$

*for $T \geq \max\left\{ \min_{l \in X} \frac{1}{p_l}, \frac{2 \log G - \log\left(M_f(1/2L + \alpha/\sqrt{p_k})\right)}{p_k} \right\}$.*

We believe the assumption on gradient bound $\|\nabla f(\cdot)\| \leq G$ is not strictly necessary and can be removed with more refined analysis. Nevertheless, the requirement on $T$ only logarithmically depends on the gradient bound $G$. In addition, we highlight that the convergence characterization in Theorem 3.3 is still token-dependent. Specifically, we can show that despite not knowing token distribution beforehand, CF-SGD can gain the same advantages that FA-SGD enjoys over SGD.

**Corollary 3.4** (Exponential Tail). *Suppose we have the same set of conditions given in Corollary 3.2, and $\sigma/\sqrt{L\left(f(\Theta^0) - f^*\right)} \leq 1$. Define $U_T$ as the set of users whose frequencies are within e-factor from the highest frequency: $U_T = \{i_n : n \leq \frac{1}{\tau}\}$, and $V_T$ similarly as $V_T = \{j_m : m \leq \frac{1}{\tau}\}$. We refer to $U_T$ as the top users, and $V_T$ as the top items.*

*Then given $|U|, |V| \geq \frac{1}{\tau}$, the proposed CF-SGD, compared to standard SGD:*

    *(1) Obtains the same rate of convergence, for the top users $U_T$ and top items $V_T$;*
    *(2) $\mathbb{E}\left\|\nabla f_{i_n}^\tau\right\|^2$ can converge faster by a factor of $\Omega\{\exp(\tau(n - |U_T|))\}$ for $i_n \in U \setminus U_T$ ;*

*(3)* $\mathbb{E}\left\|\nabla f_{j_m}^\tau\right\|^2$ *can converge faster by a factor of* $\Omega\left\{\exp\left(\tau(m - |V_T|)\right)\right\}$ *for* $j_m \in V \setminus V_T$.

**Corollary 3.5** (Polynomial Tail). *Suppose we have the same set of conditions given in Corollary 3.3, and* $\sigma/\sqrt{L\left(f(\Theta^0) - f^*\right)} \leq 1$. *Define* $U_T$ *as the set of users whose frequencies are within 2-factor from the highest frequency:* $U_T = \{i_n : n^{-\nu} \geq 1/16\}$, *and* $V_T$ *similarly as* $V_T = \{j_m : m^{-\nu} \geq 1/16\}$. *We refer to* $U_T$ *as the top users, and* $V_T$ *as the top items.*

*Then given* $|U|, |V| \geq 16^{1/\nu}$, *the FA-SGD, compared to standard SGD:*

*(1) Obtains the same rate of convergence, for the top users* $U_T$ *and top items* $V_T$;

*(2)* $\mathbb{E}\left\|\nabla f_{i_n}^\tau\right\|^2$ *can converge faster by a factor of* $\Omega\left\{\left(\frac{n}{|U_T|}\right)^\nu\right\}$ *for each* $i_n \in U \setminus U_T$;

*(3)* $\mathbb{E}\left\|\nabla f_{j_m}^\tau\right\|^2$ *can converge faster by a factor of* $\Omega\left\{\left(\frac{m}{|V_T|}\right)^\nu\right\}$ *for each* $j_m \in V \setminus V_T$.

The proofs of Corollary 3.4 and 3.5 follow similar lines as in the proofs of Corollary 3.2 and 3.3, which we defer to Appendix D

## 4 EXPERIMENTS

We conduct extensive experiments to verify the effectiveness of our proposed algorithms and our developed theories, on both publicly available benchmark recommendation datasets, and a large-scale industrial recommendation system. Additional experiments on learning Word2Vec embeddings (Mikolov et al., 2013a) are presented in Appendix B, demonstrating the general applicability of the FA/CF-SGD. We list key elements of our experiment setup for benchmark datasets below.

- **Datasets**: MovieLens-1M (GroupLens, 2003) and Criteo 1TB Click Logs dataset (Criteo, 2014).
- **Models**: Factorization Machine (FM) (Rendle, 2010), and DeepFM (Guo et al., 2017).
- **Metric**: Training loss (cross-entropy loss), and test AUC (Area Under the ROC Curve).
- **Baseline algorithms**: SGD, Adam (Kingma & Ba, 2014), Adagrad (Duchi et al., 2011). Note that Adam and Adagrad are widely popular in training recommendation systems and language models.

We also empirically verify that the token distributions for both Movielens-1M (Figure 1) and Criteo (Appendix C) dataset are highly imbalanced, with most of the token distributions having a clear polynomially or exponentially decaying tail.

Since CF-SGD does not require frequency information, which is a huge practical benefit compared to FA-SGD, in our experiments we mainly evaluate our proposed CF-SGD against the baseline algorithms. To ensure a fair comparison, for each dataset and model type, we carefully tune the learning rate of each algorithm for best performance. We apply early stopping and stop training whenever the validation AUC do not increase for 2 consecutive epochs, which is widely adopted in practice (Takács et al., 2009; Dacrema et al., 2021).

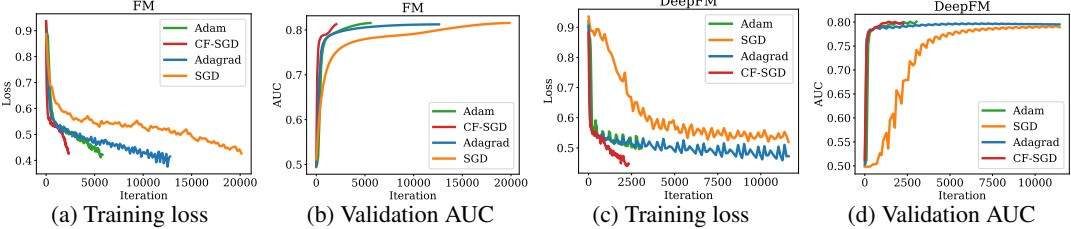

(a) Training loss      (b) Validation AUC      (c) Training loss      (d) Validation AUC

Figure 2: Movielens-1M dataset with FM and DeepFM model. CF-SGD significantly outperforms standard SGD, and is highly competitive against Adam, Adagrad.

**Movielens-1M:** We can observe from Figure 2 that for FM and DeepFM model: (1) SGD yields the slowest convergence in training loss and AUC. (2) The proposed CF-SGD yields significantly faster convergence than SGD for training loss. In addition, CF-SGD converges even faster than the adaptive learning algorithms in the early stage of training; (3) All the algorithms eventually reaches peak AUC around $81.0\%$, while CF-SGD attains the peak AUC much faster than baseline algorithms. These empirical observations help us confirm the effectiveness of the proposed CF-SGD algorithm. We further make an empirical observation that draws a close connection between adaptive algorithms and CF-SGD. We plot the second-order gradient moment maintained by Adagrad and Adam against the estimated frequency maintained by CF-SGD. Surprisingly, the second-order gradient moment quickly develops a close-to linear relationship with the frequency information accumulated by CF-SGD (Figure 3a,3b) . This observation suggests that Adagrad and Adam are exploiting frequency information implicitly to a large extent.

**Criteo:** We observe qualitative behavior of CF-SGD similar to Movielens-1M dataset, as can be seen in Figure 3c,3d, 4a,4b.

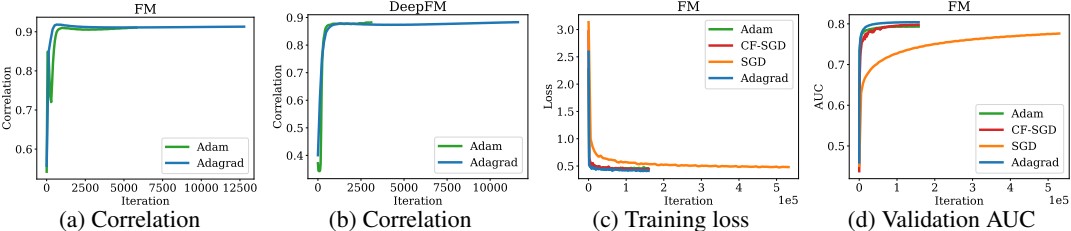

Figure 3: (a-b) Second-order gradient moment correlates linearly with frequency maintained by CF-SGD; (c-d) Comparisons on Criteo dataset with FM model.

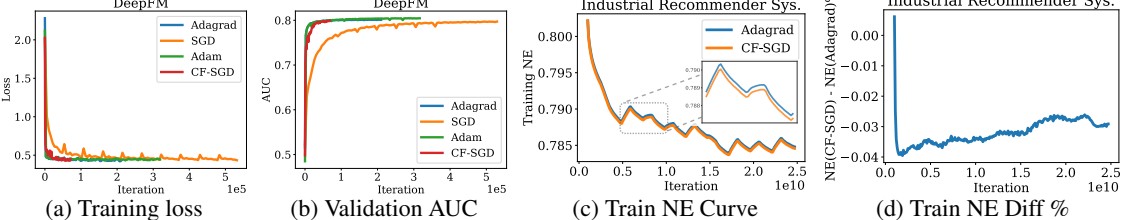

Figure 4: (a-b) Comparisons on Criteo dataset with DeepFM model; (c-d) Comparisons on a industrial-scale recommendation dataset with an ultra-large recommender model.

**Industrial Recommendation System:** We train an ultra-large industrial recommendation model with the proposed CF-SGD. The training data contains 10 days of user-item interaction records, with ~2.5 billion examples per day (*25 billion examples in total*). We use around 800 features, with ~100 million

| Alg | NE | Diff % |
|---|---|---|
| Adagrad | 0.78643 | 0.0 |
| CF-SGD | 0.78628 | -0.02 |

Table 1: Eval NE Diff %

average number of tokens per feature. We compare CF-SGD with Adagrad, which has been carefully tuned in production usage. For both algorithms, we use a batch size of 64k and do one-pass training. Different from benchmark academic datasets, we use Normalized Entropy (NE) as the evaluating metric (He et al., 2014) (smaller is better), which is the cross-entropy loss normalized by the entropy of background click through rate. Note that due to numerous iterations of the production model, any relative improvement $\sim 0.02\%$ is considered to be significant. In Figure 4c, 4d we compare the training NE curve CF-SGD and Adagrad, we can see that CF-SGD shows faster convergence than Adagrad (see NE difference % in Figure 4d). From Table 1 we can observe that CF-SGD also improves over Adagrad during the serving phase.

**Memory Efficiency:** On top of the above empirical evidences showing that CF-SGD learns *fast* – faster than standard SGD, and comparable (if not better) to adaptive algorithms, we further highlight that CF-SGD learns *cheap*. Specifically, adaptive algorithms require additional memory to store history information for each parameter. For an embedding table of size $N \times d$ ($N$ tokens, $d$ being embedding dimension), the memory needed is at least $3N \times d$ for Adam (first/second-order gradient moment), and $2N \times d$ for Adagrad (second-order gradient moment). In sharp contrast, CF-SGD only requires $N$ additional memory, for storing the estimated frequency. Since the choice of typical embedding dimension $d$ exceeds $64$ (Yin & Shen, 2018), adaptive algorithms require memory at least twice the size of the embedding table, while CF-SGD requires negligible memory overhead. Note the industrial recommendation model in our experiments has a size over multiple terabytes, with above 95% of consumed by embedding tables. Doubling the memory footprint by using standard Adam/Adagrad is infeasible in terms of both engineering and environmental concern.

## 5 CONCLUSION

We propose (Counter-based) Frequency-aware SGD for embedding learning problems, which adopts frequency-dependent learning rate schedule for each token. We demonstrate provable benefits that FA/CF-SGD enjoy over standard SGD for imbalanced token distributions, with extensive experiments supporting our theoretical findings. Our empirical findings also suggest that adaptive algorithms can implicitly exploit frequency information and hence share close connections with the proposed algorithms, this connection might be helpful in the direct analysis of adaptive algorithms for embedding learning problems, which we leave as a future direction.

ACKNOWLEDGEMENTS

We deeply appreciate Aaron Defazio and Michael Rabbat for their valuable feedbacks and insightful discussions. We are also grateful to Yuxi Hu for his help on production model experiments.

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

## A    EXPERIMENT DETAILS

### A.1    DATASETS AND PREPROCESSING

- **Movielens-1M:** Movielens-1M contains 1,000,209 anonymous ratings of approximately 3,900 movies made by 6,040 MovieLens users. Each ratings is an integer ranging from 1 to 5. We use only user id and movie id to make prediction. We treat samples with rating less or equal to 3 as negative examples, and samples with rating greater than 3 as positive examples.
- **Criteo:** The dataset consists of a portion of Criteo's traffic over a period of 7 days. Each sample corresponds to an ad served by Criteo. The label is either 0 (indicating ad being clicked) or 1 (indicating ad being ignored). The dataset consists of 13 features (integer values) and 26 categorical features. There is 45840617 total examples. For the integer valued features, we apply the log transformation ($\log(x)$ whenever $x > 2$), and convert into categorical features, as suggested by the winner of Criteo Competition [1].

For both Movielens-1M and Criteo dataset, we random split into training set, validation set and test set, taking up 80%, 10%, and 10% of the total samples respectively.

**Implementation**: We build upon torchfm[2], which contains implementation of various popular recommendation models.

### A.2    MODEL ARCHITECTURE

For all FM models, we use 64 as the embedding size. For all DeepFM models, we use 16 as the embedding size, and use (16, 16) as the widths of the hidden layers.

### A.3    HYPERPARAMETERS

For Movielens-1M dataset, the learning rate of different algorithms are list in Table 2.

| Model Type | SGD | CF-SGD | Adagrad | Adam |
|------------|-----|--------|---------|------|
| FM         | 1e1 | 1e0    | 2e-2    | 1e-3 |
| DeepFM     | 1e-2| 2e0    | 4e-2    | 2e-3 |

Table 2: Learning rates for Movielens-1M dataset.

For Criteo dataset, the learning rate of different algorithms are list in Table 3.

| Model Type | SGD | CF-SGD | Adagrad | Adam |
|------------|-----|--------|---------|------|
| FM         | 1e-2| 1e-1   | 1e-2    | 1e-3 |
| DeepFM     | 1e-2| 1e-2   | 1e-2    | 1e-3 |

Table 3: Learning rates for Criteo dataset.

All the algorithms use 1024 as the batch size during training.

## B    ADDITIONAL EXPERIMENTS ON WORD2VEC EMBEDDING LEARNING

We demonstrate the effectiveness of the proposed FA/CF-SGD for embedding learning problems in natural language modeling. Specifically, we conduct experiments for learning Word2Vec embeddings proposed in Mikolov et al. (2013a). Two learning models are considered:

**(1)** Continuous Bag-of-Words (**CBOW**): CBOW aims to predict each word (which we refer to as the center word), given its neighboring words. The training task is defined by taking each word in the corpus as the center word, and minimize the total prediction loss.

**(2)** **Skip-Gram**: Skip-Gram aims to predict each context word, given a center word. The training task is defined by taking each word in the corpus as the center word, and minimize the total prediction loss.

**Dataset and Preprocessing.** We use WikiText-2 dataset (Merity et al., 2016), which contains 36k text lines and 2M tokens in the training dataset. We remove extremely rare tokens with less than

---

[1]https://www.kaggle.com/c/criteo-display-ad-challenge/discussion/10555

[2]https://github.com/rixwew/pytorch-fm

50 occurrences in the training dataset. Note that removing extremely rare tokens was also proposed in the original Word2Vec paper Mikolov et al. (2013a), where only the top 1 million most frequent tokens are selected.

**Experiment details and results.** We choose the embedding dimension to be 300 as suggested value in Mikolov et al. (2013a). Note that for each word $w$, Word2Vec represents it by a pair of embedding vectors $(u_w, v_w)$, which we refer to as the center embedding and context embedding, respectively. Specifically, $u_w$ is used when $w$ serves as the center word, and $v_w$ is used when $w$ serves as the context word. This makes the proposed FA/CF-SGD perfectly applicable for learning Word2Vec embeddings, by simply setting $U$ as the set of center embedding vectors, and $V$ as the set of context embedding vectors.

We compare CF-SGD with standard SGD, and Adam. Following the suggestion from Mikolov et al. (2013a), we decrease the learning rate linearly as epoch increases. We use an initial stepsize of $1.0$ for both CF-SGD and SGD, and the stepsize of $0.025$ for Adam. We iterate over the training dataset for 20 epochs, with a batch size of 96. Note the original Word2Vec was trained with only 3 epochs, albeit on a much larger corpus. The results are reported in Figure 5.

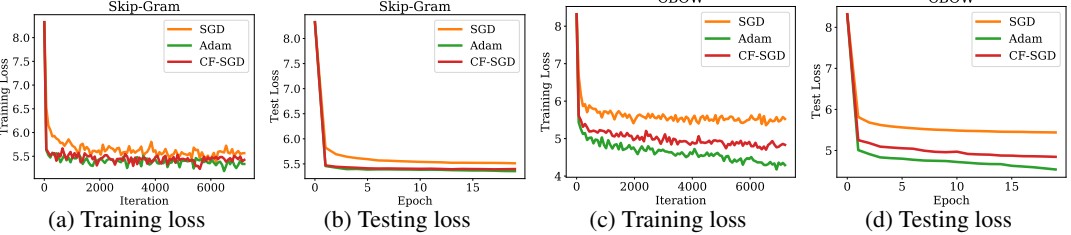

|     |     |     |     |
| :---: | :---: | :---: | :---: |
| (a) Training loss | (b) Testing loss | (c) Training loss | (d) Testing loss |

Figure 5: Comparison between CF-SGD, SGD, and Adam for learning Word2Vec embeddings on WikiText-2 dataset.

One can clearly see that for both CBOW and Skip-Gram models, CF-SGD is able to significantly improve over standard SGD. For Skip-Gram model, we observe that CF-SGD even yields comparable performance to Adam. For CBOW model, CF-SGD is able fill in the huge performance gap between Adam and SGD, and yields similar testing performance compared to Adam. Note that we do not extensively tune the initial stepsize of CF-SGD, and the linearly-decaying stepsize annealing rule was proposed in Mikolov et al. (2013a) for speeding-up SGD, which we believe might not the optimal choice for CF-SGD. We believe further improvements can be made by searching for the best initial learning rate and proper stepsize annealing rule for CF-SGD.

## C  REAL WORLD TOKEN DISTRIBUTIONS

We plot token distributions for the first 28 features (after preprocessing) of the benchmark recommendation dataset Criteo. Note that the semantic information of features for the Criteo dataset is

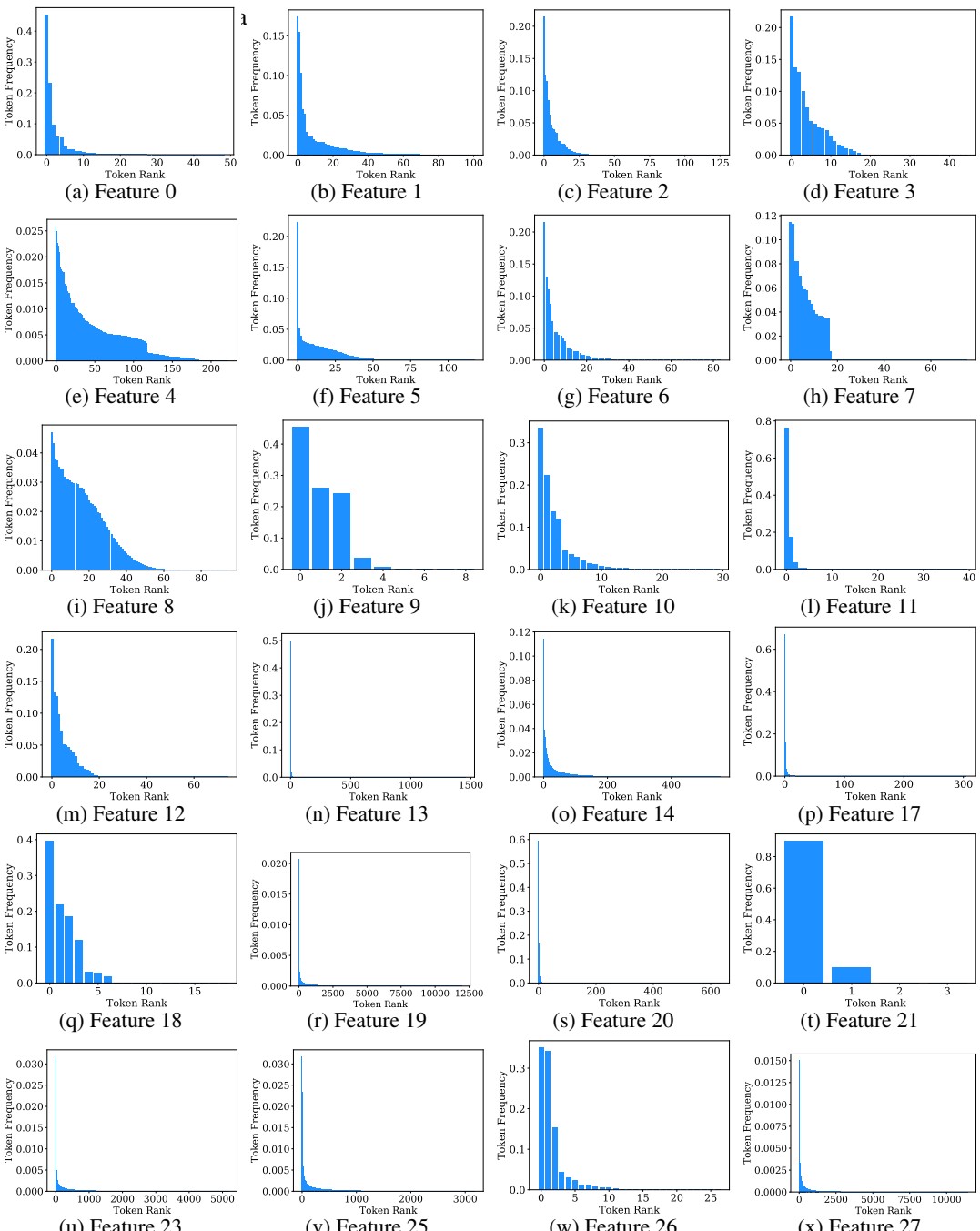

# D  ANALYSIS

Throughout our analysis, we use $\xi_t = (i_t, j_t)$ to denote the random user/item ids sampled from the unknown distribution $\mathcal{D}$. We use $\xi_{[t]} = \{\xi_s\}_{s=0}^{t-1}$ to denote the random samples collected up to the beginning of the $t$-th iteration, and use $\xi$ and $\xi_{[T]}$ interchangeably when the context is clear. Finally, we use $\mathcal{F}_t = \sigma(\xi_{[t]})$ to denote the $\sigma$-algebra generated by the random variables $\xi_{[t]}$.

*Proof of Proposition 2.1.* Let $\nabla f_U^t \in \mathbb{R}^{|U| \times d}$ denote the submatrix that contains gradient of users embeddings, and $\nabla f_V^t \in \mathbb{R}^{|V| \times d}$ for the gradient of item embeddings. Similarly, let $g_U^t \in \mathbb{R}^{|U| \times d}, g_V^t \in \mathbb{R}^{|V| \times d}$ denote the stochastic gradient of user and item embeddings, respectively.

$$\mathbb{E}_{i_t, j_t} \left\| g_U^t - \nabla f_U^t \right\|^2 = \mathbb{E}_{i_t} \left[ \mathbb{E}_{j_t | i_t} \left\| \nabla f_{i_t}^t - g_{i_t}^t \right\|^2 + \sum_{i \neq i_t, i \in U} \left\| \nabla f_i^t \right\|^2 \right]$$

$$= \mathbb{E}_{i_t} \left[ (1 - \frac{1}{p_{i_t}})^2 \left\| \nabla f_{i_t}^t \right\|^2 + \mathbb{E}_{j_t | i_t} \left\| \delta_{i_t}^t \right\|^2 + \sum_{i \neq i_t, i \in U} \left\| \nabla f_i^t \right\|^2 \right]$$

$$= \sum_{i \in U} p_i \mathbb{E}_{j_t | i_t = i} \left\| \delta_{i_t}^t \right\|^2 + \sum_{i \in U} p_i (1 - \frac{1}{p_i})^2 \left\| \nabla f_i^t \right\|^2 + \sum_{i \in U} p_i \sum_{i' \neq i} \left\| \nabla f_{i'}^t \right\|^2$$

$$= \sum_{i \in U} p_i \mathbb{E}_{j_t | i_t = i} \left\| \delta_{i_t}^t \right\|^2 + \sum_{i \in U} p_i (1 - \frac{1}{p_i})^2 \left\| \nabla f_i^t \right\|^2 + \sum_{i \in U} (1 - p_i) \left\| \nabla f_i^t \right\|^2 .$$

Similarly, we can show

$$\mathbb{E}_{i_t, j_t} \left\| g_V^t - \nabla f_V^t \right\|^2 = \sum_{j \in V} p_j \mathbb{E}_{i_t | j_t = j} \left\| \delta_{j_t}^t \right\|^2 + \sum_{j \in V} p_j (1 - \frac{1}{p_j})^2 \left\| \nabla f_j^t \right\|^2 + \sum_{j \in V} (1 - p_j) \left\| \nabla f_j^t \right\|^2 .$$

Note that $\| g^t - \nabla f^t \|^2 = \left\| g_U^t - \nabla f_U^t \right\|^2 + \left\| g_V^t - \nabla f_V^t \right\|^2$, from which we conclude the proof.  $\square$

**Proposition D.1.** *Given Assumption 2, let $\delta_i, \delta_j \in \mathbb{R}^d$, and $\Theta'$ satisfy $\theta_i' = \theta_i + \delta_i, \delta_j' = \theta_j + \delta_j$, we have*

$$f(\Theta') \leq f(\Theta) + \langle \nabla_{\theta_i} f, \delta_i \rangle + \langle \nabla_{\theta_j} f, \delta_j \rangle + \frac{\overline{L}_i}{2} \| \delta_i \|^2 + \frac{\overline{L}_j}{2} \| \delta_j \|^2 ,$$

*where $\overline{L}_k = 2 L p_k$ for all $k \in X$.*

*Proof.* Apply second-order Taylor expansion, we have

$$f(\Theta') = f(\Theta) + \langle \nabla_{\theta_i} f, \delta_i \rangle + \langle \nabla_{\theta_j} f, \delta_j \rangle + \frac{1}{2} (\delta_i^\top, \delta_j^\top) \begin{pmatrix} \widetilde{H} & \widetilde{E} \\ \widetilde{E}^\top & \widetilde{F} \end{pmatrix} (\delta_i, \delta_j)$$

where

$$\widetilde{H} = \nabla_{\theta_i \theta_i}^2 f(\widetilde{\Theta}) = \sum_{j \in V} D(i, j) \nabla_{uu}^2 \ell(\widetilde{\theta}_i, \widetilde{\theta}_j; y_{ij}),$$

$$\widetilde{E} = \nabla_{\theta_i \theta_j}^2 f(\widetilde{\Theta}) = D(i, j) \nabla_{uv} \ell(\widetilde{\theta}_i, \widetilde{\theta}_j; y_{ij}),$$

$$\widetilde{F} = \nabla_{\theta_j \theta_j}^2 f(\widetilde{\Theta}) = \sum_{i \in U} D(i, j) \nabla_{vv}^2 \ell(\widetilde{\theta}_i, \widetilde{\theta}_j; y_{ij}) = \sum_{i \in U} D(i, j) \nabla_{uu}^2 \ell(\widetilde{\theta}_i, \widetilde{\theta}_j; y_{ij}),$$

for some $\widetilde{\Theta}$ as convex combination of $\Theta$ and $\Theta'$, and the last equality uses the fact that $\ell(u; v)$ is symmetric w.r.t $u$ and $v$. Now given the assumption that $\left\| \nabla_{uu}^2 \ell(\cdot, \cdot; \cdot) \right\|_2 \leq L, \left\| \nabla_{uv}^2 \ell(\cdot, \cdot; \cdot) \right\|_2 \leq L,$

we have

$$
\begin{aligned}
f(\Theta') &= f(\Theta) + \langle \nabla_{\theta_i} f, \delta_i \rangle + \langle \nabla_{\theta_j} f, \delta_j \rangle + \delta_i^\top \widetilde{H} \delta_i + \delta_j^\top \widetilde{F} \delta_j + 2\delta_i^\top \widetilde{E} \delta_j \\
&\leq f(\Theta) + \langle \nabla_{\theta_i} f, \delta_i \rangle + \langle \nabla_{\theta_j} f, \delta_j \rangle + \frac{L}{2} \bigg( \sum_{j' \in V} D(i, j') \|\delta_i\|^2 + \sum_{u' \in U} D(i', j) \|\delta_j\|^2 \\
&\qquad + D(i, j) \|\delta_i\|^2 + D(i, j) \|\delta_j\|^2 \bigg) \\
&= f(\Theta) + \langle \nabla_{\theta_i} f, \delta_i \rangle + \langle \nabla_{\theta_j} f, \delta_j \rangle + \sum_{j' \in V} LD(i, j') \|\delta_i\|^2 + \sum_{u' \in U} LD(i', j) \|\delta_j\|^2 \\
&= f(\Theta) + \langle \nabla_{\theta_i} f, \delta_i \rangle + \langle \nabla_{\theta_j} f, \delta_j \rangle + \frac{\overline{L}_i}{2} \|\delta_i\|^2 + \frac{\overline{L}_j}{2} \|\delta_j\|^2 ,
\end{aligned}
$$

where in the first inequality we use $\delta_i^\top \widetilde{E} \delta_j \leq L \|\delta_i\| \|\delta_j\| \leq \frac{L}{2}(\|\delta_i\|^2 + \|\delta_j\|^2)$, and in the last equality we use the definition that $\overline{L}_k = 2Lp_k$ for all $k \in X$.

$\square$

Before we specify the concrete learning rate, we have the following generic convergence characterization.

**Proposition D.2.** *Given learning rate* $\{\eta_k^t\}_{k \in X, t \in [T]}$, *we have the following holds for Algorithm 1.*

$$
\mathbb{E} \sum_{t=0}^{T} \sum_{k \in X} \left( \eta_k^t - \frac{\overline{L}_k (\eta_k^t)^2}{p_k} \right) \left\| \nabla f_k^t \right\|^2 \leq f(\Theta^0) - f^* + 2 \sum_{t=0}^{T} \sum_{k \in X} p_k \overline{L}_k (\eta_k^t)^2 \sigma_k^2 .
$$

*Proof.* From Proposition D.1, we have

$$
\begin{aligned}
f(\Theta^{t+1}) &\leq f(\Theta^t) + \langle \nabla f_{i_t}^t, \theta_{i_t}^{t+1} - \theta_{i_t}^t \rangle + \langle \nabla f_{j_t}^t, \theta_{j_t}^{t+1} - \theta_{j_t}^t \rangle + \frac{\overline{L}_{i_t}}{2} \left\| \theta_{i_t}^{t+1} - \theta_{i_t}^t \right\|^2 + \frac{\overline{L}_{j_t}}{2} \left\| \theta_{j_t}^{t+1} - \theta_{j_t}^t \right\|^2 \\
&= f(\Theta^t) - \eta_{i_t}^t \langle \nabla f_{i_t}^t, g_{i_t}^t \rangle - \eta_{j_t}^t \langle \nabla f_{j_t}^t, g_{j_t}^t \rangle + \frac{\overline{L}_{i_t}}{2} (\eta_{i_t}^t)^2 \left\| g_{i_t}^t \right\|^2 + \frac{\overline{L}_{j_t}}{2} (\eta_{j_t}^t)^2 \left\| g_{j_t}^t \right\|^2 \qquad (14)
\end{aligned}
$$

Conditioned on past history $\mathcal{F}_t$, we have

$$
\mathbb{E}_{i_t, j_t} \left[ \eta_{i_t}^t \langle \nabla f_{i_t}^t, g_{i_t}^t \rangle \right] = \mathbb{E}_{i_t} \mathbb{E}_{j_t | i_t} \left[ \eta_{i_t}^t \langle \nabla f_{i_t}^t, g_{i_t}^t \rangle \right] = \sum_{i \in U} p_i \eta_i^t \frac{\left\| \nabla f_i^t \right\|^2}{p_i} = \sum_{i \in U} \eta_i^t \left\| \nabla f_i^t \right\|^2 . \quad (15)
$$

Similarly, we have

$$
\mathbb{E}_{i_t, j_t} \left[ \eta_{j_t}^t \langle \nabla f_{j_t}^t, g_{j_t}^t \rangle \right] = \sum_{i \in V} \eta_j^t \left\| \nabla f_j^t \right\|^2 . \quad (16)
$$

On the other hand, we have

$$
\begin{aligned}
\mathbb{E}_{i_t, j_t} \left[ \overline{L}_{i_t} (\eta_{i_t}^t)^2 \left\| g_{i_t}^t \right\|^2 \right] &= \mathbb{E} \left[ \overline{L}_{i_t} (\eta_{i_t}^t)^2 \left\| \frac{1}{p_{i_t}} \nabla f_{i_t}^t + \delta_{i_t}^t \right\|^2 \right] \\
&\leq 2 \mathbb{E}_{i_t} \mathbb{E}_{j_t | i_t} \left[ \overline{L}_{i_t} (\eta_{i_t}^t)^2 \left( \frac{\left\| \nabla f_{i_t}^t \right\|^2}{p_{i_t}^2} + \left\| \delta_{i_t}^t \right\|^2 \right) \right] \\
&\leq 2 \mathbb{E}_{i_t} \overline{L}_{i_t} (\eta_{i_t}^t)^2 \left( \frac{\left\| \nabla f_{i_t}^t \right\|^2}{p_{i_t}^2} + \sigma_{i_t}^2 \right) \\
&= 2 \sum_{i \in U} p_i \overline{L}_i (\eta_i^t)^2 \left( \frac{\left\| \nabla f_i^t \right\|^2}{p_i^2} + \sigma_i^2 \right) , \qquad (17)
\end{aligned}
$$

where the first inequality uses $\|a + b\|^2 \leq 2\|a\|^2 + 2\|b\|^2$, the second inequality uses (5), and the final equality uses the definition of $\overline{L}_i$ in Proposition D.1. Following similar arguments, we also have

$$\mathbb{E}_{i_t, j_t}\left[L_{i_t, j_t}(\eta_{j_t}^t)^2 \|g_{j_t}^t\|^2\right] \leq 2\sum_{j \in V} p_j \overline{L}_j(\eta_j^t)^2 \left(\frac{\|\nabla f_j^t\|^2}{p_j^2} + \sigma_j^2\right). \tag{18}$$

Plug in (15) (16) (17) (18) back into (14), we obtain

$$\mathbb{E}_{i_t, j_t}\left[f(\Theta^{t+1})|\mathcal{F}_t\right] \leq f(\Theta^t) - \sum_{i \in U}\left(\eta_i^t - \frac{\overline{L}_i(\eta_i^t)^2}{p_i}\right)\|\nabla f_i^t\|^2 - \sum_{j \in V}\left(\eta_j^t - \frac{\overline{L}_j(\eta_j^t)^2}{p_j}\right)\|\nabla f_j^t\|^2$$

$$+ 2\left(\sum_{i \in U} p_i \overline{L}_i(\eta_i^t)^2 \sigma_i^2 + \sum_{j \in U} p_j \overline{L}_j(\eta_j^t)^2 \sigma_j^2\right)$$

$$= f(\Theta^t) - \sum_{k \in X}\left(\eta_k^t - \frac{\overline{L}_k(\eta_k^t)^2}{p_k}\right)\|\nabla f_k^t\|^2 + 2\sum_{k \in X} p_k \overline{L}_k(\eta_k^t)^2 \sigma_k^2.$$

Equivalently, we have

$$\sum_{k \in X}\left(\eta_k^t - \frac{\overline{L}_k(\eta_k^t)^2}{p_k}\right)\|\nabla f_k^t\|^2 \leq f(\Theta^t) - \mathbb{E}_{i_t, j_t}\left[f(\Theta^{t+1})|\mathcal{F}_t\right] + 2\sum_{k \in X} p_k \overline{L}_k(\eta_k^t)^2 \sigma_k^2. \tag{19}$$

Sum up (19) from $t = 0$ to $T$ and take total expectation, we have

$$\mathbb{E}\sum_{t=0}^{T}\sum_{k \in X}\left(\eta_k^t - \frac{\overline{L}_k(\eta_k^t)^2}{p_k}\right)\|\nabla f_k^t\|^2 \leq f(\Theta^0) - f^* + 2\sum_{t=0}^{T}\sum_{k \in X} p_k \overline{L}_k(\eta_k^t)^2 \sigma_k^2.$$

$\square$

*Proof of Theorem 3.1.* Given Proposition D.2, suppose we use constant stepsize, i.e., $\eta_k^t = \eta_k$ for all $t \in [T]$, and sample $\tau \sim \text{Unif}([T])$, then for any $k \in X$,

$$\mathbb{E}\sum_{t=0}^{T}\left(\eta_k - \frac{\overline{L}_k(\eta_k)^2}{p_k}\right)\|\nabla f_k^t\|^2 \leq f(\Theta^0) - f^* + 2\sum_{t=0}^{T}\sum_{k \in X} p_k \overline{L}_k(\eta_k)^2 \sigma_k^2,$$

which implies that for any $k \in X$,

$$\mathbb{E}\|\nabla f_k^\tau\|^2 \leq \frac{f(\Theta^0) - f^*}{T\left(\eta_k - \frac{\overline{L}_k(\eta_k)^2}{p_k}\right)} + 2\frac{\sum_{l \in X} p_l \overline{L}_l(\eta_l)^2 \sigma_l^2}{\left(\eta_k - \frac{\overline{L}_k(\eta_k)^2}{p_k}\right)}$$

For a given $\alpha > 0$, we choose $\{\eta_k^t\}$ as the following

$$\eta_k^t = \min\left\{\frac{1}{4L}, \frac{\alpha}{\sqrt{T p_k}}\right\},$$

Combined with Proposition D.1, we have $\eta_k - \frac{\overline{L}_k(\eta_k)^2}{p_k} = \eta_k - 2L(\eta_k)^2 \geq \frac{\eta_k}{2}$, and hence

$$\mathbb{E}\|\nabla f_k^\tau\|^2 \leq \frac{2\left(f(\Theta^0) - f^*\right)}{T\eta_k} + \frac{4\sum_{l \in X} p_l \overline{L}_l(\eta_l)^2 \sigma_l^2}{\eta_k}.$$

We can bound the first term by

$$\frac{\left(f(\Theta^0) - f^*\right)}{T\eta_k} = \frac{\left(f(\Theta^0) - f^*\right)}{T}\max\left\{4L, \frac{\sqrt{T p_k}}{\alpha}\right\}$$

$$= \mathcal{O}\left\{\frac{L\left(f(\Theta^0) - f^*\right)}{T} + \frac{\sqrt{p_k}\left(f(\Theta^0) - f^*\right)}{\alpha\sqrt{T}}\right\}.$$

In addition, since

$$\sum_{l \in X} p_l \overline{L}_l \sigma_l^2 (\eta_l)^2 \leq \sum_{l \in X} p_l^2 L \sigma_l^2 \frac{\alpha^2}{T p_l} = \frac{L \sum_{l \in X} p_l \sigma_l^2 \alpha^2}{T},$$

thus we can bound the second term by

$$\frac{\sum_{l \in X} p_l \overline{L}_l (\eta_l)^2 \sigma_l^2}{\eta_k} \leq \frac{L \sum_{l \in X} p_l \sigma_l^2 \alpha^2}{T} \max \left\{ 4L, \frac{\sqrt{p_k T}}{\alpha} \right\}$$

$$= \mathcal{O} \left\{ \frac{L^2 \alpha^2 \sum_{l \in X} p_l \sigma_l^2}{T} + \frac{L \sqrt{p_k} \sum_{l \in X} p_l \sigma_l^2 \alpha}{\sqrt{T}} \right\}.$$

Thus we have

$$\mathbb{E} \|\nabla f_k^\tau\|^2 = \mathcal{O} \left\{ \frac{L \left( f(\Theta^0) - f^* \right)}{T} + \frac{\sqrt{p_k} \left( f(\Theta^0) - f^* \right)}{\alpha \sqrt{T}} + \frac{L^2 \alpha^2 \sum_{l \in X} p_l \sigma_l^2}{T} + \frac{L \sqrt{p_k} \sum_{l \in X} p_l \sigma_l^2 \alpha}{\sqrt{T}} \right\}. \tag{20}$$

By choosing $\alpha = \sqrt{\frac{f(\Theta^0) - f^*}{L \sum_{l \in X} p_l \sigma_l^2}}$, we have

$$\frac{\sqrt{p_k} \left( f(\Theta^0) - f^* \right)}{\alpha \sqrt{T}} + \frac{L \sqrt{p_k} \sum_{l \in X} p_l \sigma_l^2 \alpha}{\sqrt{T}} = \mathcal{O} \left\{ \frac{\sqrt{p_k} \sqrt{\sum_{l \in X} p_l \sigma_l^2 (f(\Theta^0) - f^*) L}}{\sqrt{T}} \right\}, \tag{21}$$

$$\frac{L^2 \alpha^2 \sum_{l \in X} p_l \sigma_l^2}{T} = \frac{L \left( f(\Theta^0) - f^* \right)}{T}. \tag{22}$$

Thus combining (20) (21) (22), we have

$$\mathbb{E} \|\nabla f_k^\tau\|^2 = \mathcal{O} \left\{ \frac{L \left( f(\Theta^0) - f^* \right)}{T} + \frac{\sqrt{p_k} \sqrt{\sum_{l \in X} p_l \sigma_l^2 (f(\Theta^0) - f^*) L}}{\sqrt{T}} \right\}, \quad \forall k \in X.$$

$$\square$$

*Proof of Theorem 3.2.* Given Proposition D.2, suppose we use token-agnostic constant stepsize, i.e., $\eta_k^t = \eta$ for all $t \in [T], k \in X$, and sample $\tau \sim \text{Unif}([T])$, then for any $k \in X$,

$$\mathbb{E} \sum_{t=0}^T \left( \eta - \frac{\overline{L}_k(\eta)^2}{p_k} \right) \|\nabla f_k^t\|^2 \leq f(\Theta^0) - f^* + 2 \sum_{t=0}^T \sum_{k \in X} p_k \overline{L}_k(\eta)^2 \sigma_k^2,$$

which implies that for any $k \in X$,

$$\mathbb{E} \|\nabla f_k^\tau\|^2 \leq \frac{f(\Theta^0) - f^*}{T \left( \eta - \frac{\overline{L}_k(\eta)^2}{p_k} \right)} + 2 \frac{\sum_{l \in X} p_l \overline{L}_l(\eta_l)^2 \sigma_l^2}{\left( \eta - \frac{\overline{L}_k(\eta)^2}{p_k} \right)}$$

For a given $\alpha > 0$, we choose $\{\eta^t\}$ as the following

$$\eta^t = \min \left\{ \frac{1}{4L}, \frac{\alpha}{\sqrt{T}} \right\},$$

Combined with Proposition D.1, we have $\eta - \frac{\overline{L}_k(\eta)^2}{p_k} = \eta - 2L(\eta)^2 \geq \frac{\eta}{2}$, and hence

$$\mathbb{E} \|\nabla f_k^\tau\|^2 \leq \frac{2 \left( f(\Theta^0) - f^* \right)}{T \eta} + \frac{4 \sum_{l \in X} p_l \overline{L}_l(\eta_l)^2 \sigma_l^2}{\eta}.$$

We can bound the first term by

$$\frac{\left(f(\Theta^0) - f^*\right)}{T\eta} = \frac{\left(f(\Theta^0) - f^*\right)}{T} \max\left\{4L, \frac{\sqrt{T}}{\alpha}\right\}$$

$$= \mathcal{O}\left\{\frac{L\left(f(\Theta^0) - f^*\right)}{T} + \frac{\left(f(\Theta^0) - f^*\right)}{\alpha\sqrt{T}}\right\}.$$

In addition, since

$$\sum_{l\in X} p_l \overline{L}_l \sigma_l^2 (\eta_l)^2 \leq \sum_{l\in X} p_l^2 L \sigma_l^2 \frac{\alpha^2}{T} = \frac{L\sum_{l\in X} p_l^2 \sigma_l^2 \alpha^2}{T},$$

thus we can bound the second term by

$$\frac{\sum_{l\in X} p_l^2 \overline{L}_l (\eta_l)^2 \sigma_l^2}{\eta} \leq \frac{L\sum_{l\in X} p_l^2 \sigma_l^2 \alpha^2}{T} \max\left\{4L, \frac{\sqrt{T}}{\alpha}\right\}$$

$$= \mathcal{O}\left\{\frac{L^2\alpha^2 \sum_{l\in X} p_l^2 \sigma_l^2}{T} + \frac{L\sum_{l\in X} p_l^2 \sigma_l^2 \alpha}{\sqrt{T}}\right\}.$$

Thus we have

$$\mathbb{E}\|\nabla f_k^\tau\|^2 = \mathcal{O}\left\{\frac{L\left(f(\Theta^0) - f^*\right)}{T} + \frac{\left(f(\Theta^0) - f^*\right)}{\alpha\sqrt{T}} + \frac{L^2\alpha^2 \sum_{l\in X} p_l^2 \sigma_l^2}{T} + \frac{L\sum_{l\in X} p_l^2 \sigma_l^2 \alpha}{\sqrt{T}}\right\}. \tag{23}$$

By choosing $\alpha = \sqrt{\frac{f(\Theta^0) - f^*}{L\sum_{l\in X} p_l^2 \sigma_l^2}}$, we have

$$\frac{\left(f(\Theta^0) - f^*\right)}{\alpha\sqrt{T}} + \frac{L\sum_{l\in X} p_l^2 \sigma_l^2 \alpha}{\sqrt{T}} = \mathcal{O}\left\{\frac{\sqrt{\sum_{l\in X} p_l^2 \sigma_l^2 (f(\Theta^0) - f^*)L}}{\sqrt{T}}\right\}, \tag{24}$$

$$\frac{L^2\alpha^2 \sum_{l\in X} p_l^2 \sigma_l^2}{T} = \frac{L\left(f(\Theta^0) - f^*\right)}{T}. \tag{25}$$

Thus combining (23) (24) (25), we have

$$\mathbb{E}\|\nabla f_k^\tau\|^2 = \mathcal{O}\left\{\frac{L\left(f(\Theta^0) - f^*\right)}{T} + \frac{\sqrt{\sum_{l\in X} p_l^2 \sigma_l^2 (f(\Theta^0) - f^*)L}}{\sqrt{T}}\right\}, \quad \forall k \in X.$$

$$\square$$

*Proof of Theorem 3.3.* We define $\widehat{p}_i^t = \sum_{s=0}^t \mathbb{1}\{i_t = i\}/t$, $\widehat{p}_j^t = \sum_{s=0}^t \mathbb{1}\{j_t = j\}/t$, and $\widehat{\eta}_k^t = \min\left\{\frac{1}{2L}, \frac{\alpha}{\sqrt{T\widehat{p}_k^t}}\right\}$ for all $t \in [T], i \in U, j \in V, k \in X$. In addition, we define the frequency-dependent learning rates $\eta_k^t = \min\left\{\frac{1}{2L}, \frac{\alpha}{\sqrt{Tp_k}}\right\}$.

Note that (19) still holds. Sum up (19) from $t = 0$ to $T$ and take total expectation, we have

$$\mathbb{E}_\xi \sum_{t=0}^T \sum_{k\in X} \left(\widehat{\eta}_k^t - \frac{\overline{L}_k (\widehat{\eta}_k^t)^2}{p_k}\right) \|\nabla f_k^t\|^2 \leq f(\Theta^0) - f^* + 2\mathbb{E}_\xi \sum_{t=0}^T \sum_{k\in X} p_k \overline{L}_k (\widehat{\eta}_k^t)^2 \sigma_k^2. \tag{26}$$

In contrast to FA-SGD and standard SGD, here the stepsize $\widehat{\eta}_k^t \in \mathcal{F}_t$ is also a random variable. We first proceed to upper bound the right hand side of (26).

For each $k \in X$, denote $T_k = \frac{c}{p_k}$, where $c > 0$ is any absolute constant. Note that we have for any $t \geq T_k$,

$$\mathbb{P}\left(|\widehat{p}_k^t - p_k| \geq \frac{p_k}{2}\right) \leq \mathbb{P}\left(|\widehat{p}_k^t - p_k| \geq \frac{p_k}{2}\right) \leq \exp\left(-\frac{tp_k^2}{p_k(1-p_k)}\right) \leq \exp\left(-tp_k\right). \quad (27)$$

Moreover, we have

$$|\widehat{p}_k^t - p_k| \leq \frac{p_k}{2} \quad \Rightarrow \quad |\widehat{\eta}_k^t - \eta_k^t| \leq \alpha_0 \eta_k^t, \quad (28)$$

where $\alpha_0 = \max\left\{1 - \sqrt{2/3}, \sqrt{2} - 1\right\} < 1$. Note that for any $T > T_k$,

$$\sum_{t=0}^{T} p_k \overline{L}_k \sigma_k^2 \mathbb{E}_\xi\left[(\widehat{\eta}_k^t)^2\right] = \sum_{t=0}^{T_k} p_k \overline{L}_k \sigma_k^2 \mathbb{E}_\xi\left[(\widehat{\eta}_k^t)^2\right] + \sum_{t=T_k+1}^{T} p_k \overline{L}_k \sigma_k^2 \mathbb{E}_\xi\left[(\widehat{\eta}_k^t)^2\right],$$

To bound the first term, note that from definition of $\widehat{\eta}_k^t$, we have $\widehat{\eta}_k^t \leq \frac{1}{L}$, hence

$$\sum_{t=0}^{T_k} p_k \overline{L}_k \sigma_k^2 \mathbb{E}_\xi\left[(\widehat{\eta}_k^t)^2\right] \leq \sum_{t=0}^{T_k} p_k \overline{L}_k \sigma_k^2 \left(\frac{1}{L}\right)^2 = T_k p_k \overline{L}_k \sigma_k^2 \left(\frac{1}{L}\right)^2 = c\overline{L}_k \sigma_k^2 \left(\frac{1}{L}\right)^2.$$

To bound the second term, note that from (27), for any $t \geq T_k$ with probability at least $1 - \delta_k^t$ (here $\delta_k^t = \exp(-tp_k)$), we have that $|\eta_k^t - \widehat{\eta}_k^t| \leq \alpha_0 \eta_k^t$. Denote $\mathcal{H}_k^t = \{\omega : |\eta_k^t - \widehat{\eta}_k^t| \leq \alpha_0 \eta_k^t\}$, we have

$$\mathbb{E}_\xi(\widehat{\eta}_k^t)^2 = \mathbb{E}_\xi(\widehat{\eta}_k^t)^2 \mathbb{1}_{\mathcal{H}_k^t} + \mathbb{E}_\xi(\widehat{\eta}_k^t)^2 \mathbb{1}_{(\mathcal{H}_k^t)^c}$$

$$\leq (1+\alpha_0)^2 (\eta_k^t)^2 \mathbb{E}_\xi \mathbb{1}_{\mathcal{H}_k^t} + \left(\frac{1}{L}\right)^2 \mathbb{E}_\xi \mathbb{1}_{(\mathcal{H}_k^t)^c}$$

$$\leq (1+\alpha_0)^2 (\eta_k^t)^2 + \delta_k^t \left(\frac{1}{L}\right)^2.$$

Hence for any $t \geq T_k$,

$$\sum_{t=T_k+1}^{T} p_k \overline{L}_k \sigma_k^2 \mathbb{E}_\xi\left[(\widehat{\eta}_k^t)^2\right] \leq \sum_{t=0}^{T} (1+\alpha_0)^2 p_k \overline{L}_k \sigma_k^2 (\eta_k^t)^2 + \sum_{t=T_k+1}^{T} \delta_k^t p_k \overline{L}_k \sigma_k^2 \left(\frac{1}{L}\right)^2.$$

Thus we obtain

$$\sum_{t=0}^{T} p_k \overline{L}_k \sigma_k^2 \mathbb{E}_\xi\left[(\widehat{\eta}_k^t)^2\right] \leq c\overline{L}_k \sigma_k^2 \left(\frac{1}{L}\right)^2 + \sum_{t=0}^{T} (1+\alpha_0)^2 p_k \overline{L}_k \sigma_k^2 (\eta_k^t)^2 + \sum_{t=T_k+1}^{T} \delta_k^t p_k \overline{L}_k \sigma_k^2 \left(\frac{1}{L}\right)^2$$

$$\leq c\overline{L}_k \sigma_k^2 \left(\frac{1}{L}\right)^2 + \sum_{t=0}^{T} (1+\alpha_0)^2 p_k \overline{L}_k \sigma_k^2 (\eta_k^t)^2 + \frac{\exp(-c)p_k}{1-\exp(-p_k)} \overline{L}_k \sigma_k^2 \left(\frac{1}{L}\right)^2$$

$$\leq c\overline{L}_k \sigma_k^2 \left(\frac{1}{L}\right)^2 + \sum_{t=0}^{T} (1+\alpha_0)^2 p_k \overline{L}_k \sigma_k^2 (\eta_k^t)^2 + \alpha_1 \exp(-c) \overline{L}_k \sigma_k^2 \left(\frac{1}{L}\right)^2,$$

where the last inequality uses the fact that $\sum_{t=T_k+1}^{T} \delta_k^t \leq \sum_{t=T_k}^{\infty} \delta_k^t = \frac{\exp(-T_k p_k)}{1-\exp(-p_k)} = \frac{\exp(-c)}{1-\exp(-p_k)}$, and $\alpha_1 = \sup_{p \in (0,1)} \frac{p}{1-\exp(-p)}$.

Hence by denoting $T_0 = \max_{k \in X} T_k$, we have that for any $k \in X$, and $t \geq T_0$,

$$\mathbb{E}_\xi \sum_{t=0}^{T} \left(\widehat{\eta}_k^t - \frac{\overline{L}_k (\widehat{\eta}_k^t)^2}{p_k}\right) \left\|\nabla f_k^t\right\|^2 \leq f(\Theta^0) - f^* + 2\Bigg\{ \underbrace{\sum_{k \in X} c\overline{L}_k \sigma_k^2 \left(\frac{1}{L}\right)^2}_{(A)}$$

$$+ \underbrace{\sum_{t=0}^{T} \sum_{k \in X} (1+\alpha_0)^2 p_k \overline{L}_k \sigma_k^2 (\eta_k^t)^2}_{(B)} + \underbrace{\alpha_1 \exp(-c) \overline{L}_k \sigma_k^2 \left(\frac{1}{L}\right)^2}_{(C)} \Bigg\}.$$

$$(29)$$

Note that term (B) in (29) can be bounded following exactly the same step as in the proof of Theorem 3.1, for which we have

$$T \sum_{l \in X} p_l \overline{L}_l \sigma_l^2 (\eta_l)^2 \leq T \sum_{l \in X} p_l^2 L \sigma_l^2 \frac{\alpha^2}{T p_l} = L \sum_{l \in X} p_l \sigma_l^2 \alpha^2.$$

Hence for any constant $c > 0$ (we can readily choose $c = 1$), we have

$$\mathbb{E}_\xi \sum_{t=0}^{T} \left( \widehat{\eta}_k^t - \frac{\overline{L}_k (\widehat{\eta}_k^t)^2}{p_k} \right) \left\| \nabla f_k^t \right\|^2 = f(\Theta^0) - f^* + 2 \Bigg\{ \underbrace{c \sum_{k \in X} \overline{L}_k \sigma_k^2 \left( \frac{1}{L} \right)^2}_{(A')}$$

$$+ \underbrace{(1 + \alpha_0)^2 L \sum_{k \in X} p_k \sigma_k^2 \alpha^2}_{(B')} + \underbrace{\sum_{k \in X} \alpha_1 \exp(-c) \overline{L}_k \sigma_k^2 \left( \frac{1}{L} \right)^2}_{(C')} \Bigg\}.$$

By the definition of $\widehat{\eta}_k^t$, we also have $\widehat{\eta}_k^t - \frac{\overline{L}_k (\widehat{\eta}_k^t)^2}{p_k} \geq \frac{\widehat{\eta}_k^t}{2}$, then

$$\mathbb{E}_\xi \sum_{t=0}^{T} \left( \widehat{\eta}_k^t - \frac{\overline{L}_k (\widehat{\eta}_k^t)^2}{p_k} \right) \left\| \nabla f_k^t \right\|^2 \geq \mathbb{E}_\xi \sum_{t=0}^{T} \frac{\widehat{\eta}_k^t}{2} \left\| \nabla f_k^t \right\|^2.$$

Hence we obtain

$$\mathbb{E}_\xi \sum_{t=0}^{T} \frac{\widehat{\eta}_k^t}{2} \left\| \nabla f_k^t \right\|^2 \leq f(\Theta^0) - f^* + 2 \Bigg\{ \underbrace{c \sum_{k \in X} \overline{L}_k \sigma_k^2 \left( \frac{1}{L} \right)^2}_{(A')}$$

$$+ \underbrace{(1 + \alpha_0)^2 L \sum_{k \in X} p_k \sigma_k^2 \alpha^2}_{(B')} + \underbrace{\sum_{k \in X} \alpha_1 \exp(-c) \overline{L}_k \sigma_k^2 \left( \frac{1}{L} \right)^2}_{(C')} \Bigg\}, \quad (30)$$

or equivalently,

$$\mathbb{E}_\xi \sum_{t=0}^{T} \frac{\widehat{\eta}_k^t}{\sum_{t=0}^{T} \eta_k^t} \left\| \nabla f_k^t \right\|^2 \leq \frac{2 \left( f(\Theta^0) - f^* \right)}{\sum_{t=0}^{T} \eta_k^t} + \frac{4(A')}{\sum_{t=0}^{T} \eta_k^t} + \frac{4(B')}{\sum_{t=0}^{T} \eta_k^t} + \frac{4(C')}{\sum_{t=0}^{T} \eta_k^t}. \quad (31)$$

Let $\widetilde{T}_0$ denote a positive integer to be determined later, recall that from (27), for any $t \geq T_0$, with probability at least $1 - \widetilde{\delta}_k^t$ (here $\widetilde{\delta}_k^t = \exp(-t p_k)$), we have $|\eta_k^t - \widehat{\eta}_k^t| \leq \alpha_0 \eta_k^t$ hold. Denote $\mathcal{B}_k^t = \{ w : |\eta_k^t - \widehat{\eta}_k^t| \leq \alpha_0 \eta_k^t \}$, then we have that for any $k \in X$,

$$\mathbb{E}_\xi \sum_{t=0}^{T} \frac{\widehat{\eta}_k^t}{\sum_{t=0}^{T} \eta_k^t} \left\| \nabla f_k^t \right\|^2 \geq \mathbb{E}_\xi \sum_{t=\widetilde{T}_0}^{T} \frac{\widehat{\eta}_k^t}{\sum_{t=0}^{T} \eta_k^t} \left\| \nabla f_k^t \right\|^2$$

$$\geq \mathbb{E}_\xi \sum_{t=\widetilde{T}_0}^{T} \frac{\widehat{\eta}_k^t}{\sum_{t=0}^{T} \eta_k^t} \left\| \nabla f_k^t \right\|^2 \mathbb{1}_{\mathcal{B}_k^t}$$

$$\geq (1 - \alpha_0) \sum_{t=\widetilde{T}_0}^{T} \frac{\eta_k^t}{\sum_{t=0}^{T} \eta_k^t} \mathbb{E}_\xi \left\| \nabla f_k^t \right\|^2 \mathbb{1}_{\mathcal{B}_k^t}$$

$$\geq (1 - \alpha_0) \sum_{t=\widetilde{T}_0}^{T} \frac{\eta_k^t}{\sum_{t=0}^{T} \eta_k^t} \left\{ \mathbb{E}_\xi \left\| \nabla f_k^t \right\|^2 - \widetilde{\delta}_k^t G^2 \right\}$$

$$= \frac{(1 - \alpha_0)(T - \widetilde{T}_0)}{T} \sum_{t=\widetilde{T}_0}^{T} \frac{1}{T_0 - \widetilde{T}_0} \left\{ \mathbb{E}_\xi \left\| \nabla f_k^t \right\|^2 - \widetilde{\delta}_k^t G^2 \right\}$$

$$\geq \frac{(1 - \alpha_0)(T - \widetilde{T}_0)}{T} \left\{ \mathbb{E}_\tau \mathbb{E}_\xi \left\| \nabla f_k^\tau \right\|^2 - \widetilde{\delta}_k^{T_0} G^2 \right\}, \quad (32)$$

where the fourth inequality uses the assumption that $\|\nabla f_k^t\|^2 \leq G^2$, and the last equality follows from the definition that $\tau \sim \text{Unif}\{\widetilde{T}_0, \ldots, T\}$. Finally, choose $\widetilde{T}_0 = T/2$, combine (31) and (32), we obtain

$$\mathbb{E}_\tau \mathbb{E}_\xi \|\nabla f_k^\tau\|^2 = \mathcal{O}\left\{\frac{2\left(f(\Theta^0) - f^*\right)}{\sum_{t=0}^T \eta_k^t} + \frac{4(\text{A}')}{\sum_{t=0}^T \eta_k^t} + \frac{4(\text{B}')}{\sum_{t=0}^T \eta_k^t} + \frac{4(\text{C}')}{\sum_{t=0}^T \eta_k^t} + \widetilde{\delta}_k^{T_0} G^2\right\}.$$

Combine with the definition of (A'), (B'), (C') in (30), and the definition of $\widetilde{\delta}_k^{T_0}$, we obtain

$$\mathbb{E}_\tau \mathbb{E}_\xi \|\nabla f_k^\tau\|^2 = \mathcal{O}\left\{\frac{f(\Theta^0) - f^* + \sum_{k \in X} \overline{L}_k \sigma_k^2 \left(\frac{1}{L}\right)^2}{\sum_{t=0}^T \eta_k^t} + \frac{(1+\alpha_0)^2 L \sum_{k \in X} p_k \sigma_k^2 \alpha^2}{\sum_{t=0}^T \eta_k^t} + \widetilde{\delta}_k^{T_0} G^2\right\}$$

$$= \mathcal{O}\left\{\frac{M_f}{\sum_{t=0}^T \eta_k^t} + \frac{L \sum_{k \in X} p_k \sigma_k^2 \alpha^2}{\sum_{t=0}^T \eta_k^t} + \exp(-Tp_k/2)G^2\right\}$$

$$= \mathcal{O}\left\{\frac{M_f}{\sum_{t=0}^T \eta_k^t} + \frac{L \sum_{k \in X} p_k \sigma_k^2 \alpha^2}{\sum_{t=0}^T \eta_k^t}\right\},$$

where the last inequality holds whenever $T \geq \widehat{T}_0$ and $\widehat{T}_0$ is large enough so that $\exp(-\widehat{T}_0 p_k/2)G^2 \leq \frac{M_f}{\sum_{t=0}^T \eta_k^t} \leq \frac{M_f}{\sum_{t=0}^{\widehat{T}_0} \eta_k^t}$, and the second equality follows from the definition of $M_f = f(\Theta^0) - f^* + \sum_{k \in X} \overline{L}_k \sigma_k^2 \left(\frac{1}{L}\right)^2 = f(\Theta^0) - f^* + \sum_{k \in X} p_k \sigma_k^2/L$. Finally, following similar lines as in the proof of Theorem 3.1, by choosing $\alpha = \sqrt{\frac{M_f}{L \sum_{l \in X} p_l \sigma_l^2}}$ we obtain that for $T \geq \max\left\{T_0, \widehat{T}_0\right\}$,

$$\mathbb{E}_\xi \|\nabla f_k^\tau\|^2 = \mathcal{O}\left\{\frac{LM_f}{T} + \frac{\sqrt{p_k}\sqrt{\sum_{l \in X} p_l \sigma_l^2 LM_f}}{\sqrt{T}}\right\}$$

$$= \mathcal{O}\left\{\frac{LM_f}{T} + \frac{\sqrt{p_k}\sqrt{\sum_{l \in X} p_l \sigma_l^2 L\left(f(\Theta^0) - f^*\right) + \left(\sum_{l \in X} p_l \sigma_l^2\right)^2}}{\sqrt{T}}\right\}$$

$$= \mathcal{O}\left\{\frac{LM_f}{T} + \frac{\sqrt{p_k}\sqrt{\sum_{l \in X} p_l \sigma_l^2 L\left(f(\Theta^0) - f^*\right)}}{\sqrt{T}} + \frac{\sqrt{p_k}\left(\sum_{l \in X} p_l \sigma_l^2\right)}{\sqrt{T}}\right\}, \quad \forall k \in X,$$

where the last inequality uses simple fact $\sqrt{a+b} \leq \sqrt{a} + \sqrt{b}$ whenever $a, b > 0$. It remains to estimate the order of $\widehat{T}_0$, we need $\exp(-\widehat{T}_0 p_k/2)G^2 \leq \frac{M_f}{\sum_{t=0}^{\widehat{T}_0} \eta_k^t}$, this can be readily satisfied by taking $\widehat{T}_0 = \frac{2\log G - \log\left(M_f(1/2L + \alpha/\sqrt{p_k})\right)}{p_k}$.

$\square$

*Improvement of CF-SGD over SGD.* Now under the the assumption that $\sigma_l^2 = \sigma^2$ for all $l \in X$, compared with the rate of convergence of standard SGD in (9), the improvement of Counter-based Frequency-aware SGD is governed by the ratio $\gamma_c$ defined by

$$\gamma_c := \frac{\sqrt{p_k}}{\sqrt{\sum_{l \in X} p_l^2}}\left(1 + \frac{\sigma}{\sqrt{L(f(\Theta^0) - f^*)}}\right),$$

which is only a constant factor away from the the improvement ratio $\gamma_f$ of vanilla frequency-aware SGD over standard SGD, given by

$$\gamma_f := \frac{\sqrt{p_k}}{\sqrt{\sum_{l \in X} p_l^2}}.$$

Hence both Corollary 3.2 and 3.3 hold for Counter-based Frequency-aware SGD after properly adjusting the constant factor in the statement. $\square$

*Proof of Corollary 3.1.* By trivial verification. □

**Corollary D.1** (Corollary 3.2, restated). *Let* $U = \{i_n\}_{n=1}^{|U|}$, $V = \{j_m\}_{m=1}^{|V|}$, *where* $i_n$ *denote the user with $n$-th largest frequency, and $j_m$ denote the item with the $m$-th largest frequency. Suppose*

$$p_{i_n} \propto \exp(-\tau n), \;\; p_{j_m} \propto \exp(-\tau m), \quad \forall n \in [|U|], m \in [|V|]. \tag{33}$$

*for some $\tau > 0$. Then there exists $\beta_U, \beta_V > 0$ such that for any user $i_n \in U$, we have*

$$\frac{p_{i_n} \sum_{l \in X} p_l \sigma_l^2}{\sum_{l \in X} p_l^2 \sigma_l^2} = \beta_U(\tau) \exp\left(-\tau n\right) \le \frac{1 + \exp(-\tau)}{1 - \exp(-\tau)} \exp\left(-\tau n\right). \tag{34}$$

*Similarly, for any item $j_m \in V$, we have*

$$\frac{p_{j_m} \sum_{l \in X} p_l \sigma_l^2}{\sum_{l \in X} p_l^2 \sigma_l^2} = \beta_V(\tau) \exp\left\{-\tau m\right\} \le \frac{1 + \exp(-\tau)}{1 - \exp(-\tau)} \exp\left\{-\tau m\right\}. \tag{35}$$

*In addition, there exists an absolute constant $C > 0$, such that for whenver $|U|, |V| \ge \frac{1}{\tau}$, we have $\beta_U(\tau), \beta_V(\tau) \le C$, and thus*

$$\frac{p_{i_n} \sum_{l \in X} p_l \sigma_l^2}{\sum_{l \in X} p_l^2 \sigma_l^2} \le C \exp\left(-\tau n\right); \; \frac{p_{j_m} \sum_{l \in X} p_l \sigma_l^2}{\sum_{l \in X} p_l^2 \sigma_l^2} \le C \exp\left\{-\tau m\right\}, \;\; \forall i_n \in U, \forall j_m \in V. \tag{36}$$

*Proof.* Define $M_U = \frac{1 - \exp(-\tau)}{1 - \exp(-\tau|U|)}$, and $M_V = \frac{1 - \exp(-\tau)}{1 - \exp(-\tau|V|)}$, from (33) we have $p_{i_n} = M_U \exp(-\tau n), p_{j_m} = M_V \exp(-\tau m)$. In addition, we denote

$$M_{U,V} = \sum_{l \in X} p_l^2 = \sum_{i \in U} p_i^2 + \sum_{j \in V} p_j^2 = M_U^2 \frac{1 - \exp(-2\tau|U|)}{1 - \exp(-2\tau)} + M_V^2 \frac{1 - \exp(-2\tau|U|)}{1 - \exp(-2\tau)}$$

$$= \frac{(1 + \exp(-\tau|U|))(1 - \exp(-\tau))}{(1 - \exp(-\tau|U|))(1 + \exp(-\tau))} + \frac{(1 + \exp(-\tau|V|))(1 - \exp(-\tau))}{(1 - \exp(-\tau|V|))(1 + \exp(-\tau))}.$$

Thus we obtain,

$$\frac{p_{i_n} \sum_{l \in X} p_l \sigma_l^2}{\sum_{l \in X} p_l^2 \sigma_l^2} = \frac{2p_{i_n}}{\sum_{l \in X} p_l^2} = \frac{2M_U}{M_{U,V}} \exp(-\tau n) = \beta_U \exp(-\tau n), \;\; \forall i_n \in U$$

$$\frac{p_{j_m} \sum_{l \in X} p_l \sigma_l^2}{\sum_{l \in X} p_l^2 \sigma_l^2} = \frac{2p_{j_m}}{\sum_{l \in X} p_l^2} = \frac{2M_V}{M_{U,V}} \exp(-\tau m) = \beta_V \exp(-\tau m), \;\; \forall j_m \in V,$$

From the definition of $\beta_U = \frac{2M_U}{M_{U,V}}, \beta_V = \frac{2M_V}{M_{U,V}}$, combined with the fact that $M_{U,V} \ge 2\frac{1 - \exp(-\tau)}{1 + \exp(-\tau)}$, and $M_U, M_V \le 1$, we obtain the inequality in (34) and (35). Finally, whenever $|U|, |V| \ge \frac{1}{\tau}$, we have $\beta_U \le \frac{1 - \exp(-\tau)}{1 - \exp(-\tau|U|)} \frac{1 + \exp(-\tau)}{1 - \exp(-\tau)} \le \frac{2}{1 - e^{-1}}$ and similarly $\beta_V \le \frac{2}{1 - e^{-1}}$. Let $C = \frac{2}{1 - e^{-1}}$, we obtain (36).

□

**Corollary D.2** (Corollary 3.3, restated). *Let* $U = \{i_n\}_{n=1}^{|U|}$, $V = \{j_m\}_{m=1}^{|V|}$, *where* $i_n$ *denote the user with $n$-th largest frequency, and $j_m$ denote the item with the $m$-th largest frequency. Suppose*

$$p_{i_n} \propto n^{-\nu}, \;\; p_{j_m} \propto m^{-\nu}, \quad \forall n \in [|U|], m \in [|V|]. \tag{37}$$

*for some $\nu \ge 2$. Define $U_T$ as the set of users whose frequencies are within 2-factor from the highest frequency: $U_T = \{i_n : n^{-\nu} \ge 1/2\}$, and $V_T$ similarly as $V_T = \{j_m : m^{-\nu} \ge 1/2\}$. We refer to $U_T$ as the top users, and $V_T$ as the top items. Then there exists an absolute constant $C > 0$, such that*

$$\frac{p_{i_n} \sum_{l \in X} p_l \sigma_l^2}{\sum_{l \in X} p_l^2 \sigma_l^2} \le C n^{-\nu}; \; \frac{p_{j_m} \sum_{l \in X} p_l \sigma_l^2}{\sum_{l \in X} p_l^2 \sigma_l^2} \le C m^{-\nu}, \;\; \forall i_n \in U, \forall j_m \in V.$$

*Proof.* We have $p_{i_n} = M_U n^{-\nu}$, where $M_U = \frac{1}{\sum_{n=1}^{|U|} n^{-\nu}}$. Similarly, we have $p_{j_m} = M_V m^{-\nu}$, where $M_V = \frac{1}{\sum_{m=1}^{|V|} m^{\nu}}$. Hence

$$\frac{p_{i_n} \sum_{l \in X} p_l \sigma_l^2}{\sum_{l \in X} p_l^2 \sigma_l^2} = \frac{2p_{i_n}}{\sum_{l \in X} p_l^2} = \frac{2M_U n^{-\nu}}{M_U^2 \sum_{\widetilde{n}=1}^{|U|} \widetilde{n}^{-2\nu} + M_V^2 \sum_{\widetilde{m}=1}^{|V|} \widetilde{m}^{-2\nu}}$$

We have

$$\sum_{n=1}^{|U|} n^{-\nu} = 1 + \sum_{2}^{|U|} n^{-\nu} \geq 1 + \int_2^{|U|+1} n^{-\nu} = 1 + \frac{1}{\nu-1}\left[2^{1-\nu} - (|U|+1)^{1-\nu}\right],$$

$$\sum_{n=1}^{|U|} n^{-\nu} < 1 + \int_1^{|U|+1} n^{-\nu} = 1 + \frac{1}{\nu-1}\left[1 - (|U|+1)^{1-\nu}\right],$$

$$\sum_{n=1}^{|U|} n^{-2\nu} > 1 + \frac{1}{2\nu-1}\left[2^{1-2\nu} - (|U|+1)^{1-2\nu}\right],$$

which implies

$$M_U < \frac{\nu-1}{2^{1-\nu} - (|U|+1)^{1-\nu} + \nu - 1}, \quad M_U > \frac{\nu-1}{(|U|+1)^{1-\nu} + \nu},$$

$$M_V < \frac{\nu-1}{2^{1-\nu} - (|V|+1)^{1-\nu} + \nu - 1}, \quad M_U > \frac{\nu-1}{(|V|+1)^{1-\nu} + \nu},$$

Thus

$$\frac{p_{i_n} \sum_{l \in X} p_l \sigma_l^2}{\sum_{l \in X} p_l^2 \sigma_l^2}$$

$$\leq \frac{n^{-\nu} \frac{2(2\nu-1)}{(2^{1-\nu} - (|U|+1)^{1-\nu} + \nu - 1)(\nu-1)}}{\left(\frac{1}{(|U|+1)^{1-\nu}+\nu}\right)^2 (2^{1-2\nu} - (|U|+1)^{1-2\nu} + 2\nu - 1) + \left(\frac{1}{(|V|+1)^{1-\nu}+\nu}\right)^2 (2^{1-2\nu} - (|V|+1)^{1-2\nu} + 2\nu - 1)},$$

Note that

$$2^{1-2\nu} - (|U|+1)^{1-2\nu} + 2\nu - 1 > 2\nu - 1, \quad 2^{1-2\nu} - (|V|+1)^{1-2\nu} + 2\nu - 1 > 2\nu - 1,$$

$$\frac{1}{(|U|+1)^{1-\nu}+\nu} > \frac{1}{1+\nu}, \quad \frac{1}{(|V|+1)^{1-\nu}+\nu} > \frac{1}{1+\nu},$$

$$\frac{1}{2^{1-\nu} - (|U|+1)^{1-\nu} + \nu - 1} < \frac{1}{\nu-1}, \quad \frac{1}{2^{1-\nu} - (|V|+1)^{1-\nu} + \nu - 1} < \frac{1}{\nu-1}.$$

Then we conclude with

$$\frac{p_{i_n} \sum_{l \in X} p_l \sigma_l^2}{\sum_{l \in X} p_l^2 \sigma_l^2} \leq \frac{(2\nu-1)}{(\nu-1)} \cdot \frac{(1+\nu)^2}{(\nu-1)(2\nu-1)} n^{-\nu} \leq 16 n^{-\nu},$$

where the last inequality follows from the fact that $\nu \geq 2$. Similarly we can show that

$$\frac{p_{j_m} \sum_{l \in X} p_l \sigma_l^2}{\sum_{l \in X} p_l^2 \sigma_l^2} \leq 16 m^{-\nu}.$$

Take $C = 16$, we obtain the desired result.

$\square$

