# OpenReview forum: "Frequency-aware SGD for Efficient Embedding Learning with Provable Benefits"
_ICLR.cc/2022/Conference — ICLR 2022 Poster_

### Official Review · Reviewer_Xi3e · 2021-10-31

**Correctness:** 3
**Technical Novelty And Significance:** 3
**Empirical Novelty And Significance:** 3
**Recommendation:** 6
**Confidence:** 3

**Main Review:**

Strengths
1. The proposed algorithms are easy to implement.
2. Both the online and offline situations are considered.

Weaknesses
The experiments are insufficient. Offline training is also regularly (daily or hourly) required and conducted in industrial recommender systems for capturing long-term interests. It is unknown if FA-SGD outperforms the baselines in offline evaluation. Besides, matching and ranking are two stages in recommender systems. FM and DeepFM are commonly used models in the stage of ranking. It is unknown if CF-SGD shows advantages in the stage of matching. I suggest the authors employ the algorithm to some common matching models such as DSSM for a comprehensive evaluation. In addition, only the cross-entropy (point-wise) loss is tested in the experiments. It is unknown if the proposed algorithms can generalize to adapt to other losses such as the pair-wise loss.


**Summary Of The Paper:**

This paper proposes two optimization algorithms for recommendation where the token distributions are highly imbalanced. The frequency information is integrated into the optimization algorithms for fast convergence and better performance. The proposed algorithms are easy to understand and implement, and the theoretical analysis for the bounds are provided.

**Summary Of The Review:**

This paper proposes to integrate the frequency information of tokens into the optimization algorithms for a fast convergence. The idea is novel and the proposed algorithms are easy-to-implement. I recommend to accept it. However, the experiments are not comprehensive and I want to see a thorough evaluation which considers the offline training and the matching stage. Besides, more loss functions should also be investigated.

---

> ### Author Response · Authors · 2021-11-11
> **Response to Reviewer Xi3e**
>
> We appreciate your positive feedback and valuable comments. Below we provide our response to your detailed remarks, which we hope can address your current concerns.
>
> All our academic datasets can be viewed as using offline training. That is, we do multi-pass training on the training dataset, and then evaluate the trained model on the evaluation dataset. Typical production recommendation model first uses a large amount of data (e.g., 10 days of accumulated user-item interaction) to train a model (we refer to this stage as offline training), and then conducts recurrent training using newly collected data on an hourly/daily basis (we refer to this stage as the online training), while deploying the trained model episodically. *Our algorithm is agnostic to offline and online training and works for both*, as long as the frequency information is aggregated correctly. For instance, for our production model we train with the last 10 days of data, and evaluate on the 11th day of data, the training curves are reported in Figure 4c, 4d, and the evaluation result are presented in Table 1.
>
> We gently reiterate that we view our theoretical developments as core contributions, which are completely new in the literature. The results we obtain hold beyond the scope of learning recommendation system, and the provable advantage of proposed FA/CF-SGD holds for any embedding learning task as long as the token distribution is imbalanced, regardless of network architecture or loss functions.
>
> The experiments consist of two widely popular academic datasets: Movielens-1M and Criteo. The experiments on the production recommendation model are already extremely large-scale: *multiple terabytes of model size*,  *25 billion examples* of training data, and  *2.5 billion examples* of evaluation data. The benefit of CF-SGD can already be seen even though we do not extensively tune the hyper-parameter of CF-SGD, demonstrating the robustness of the algorithm.
>
> We view the current experiments as the empirical validation of our developed theories instead of our core contribution. The core message we aim to deliver through this paper is not to develop an empirically better method for learning recommendation systems, but a new perspective on efficient algorithmic design that draws (theoretically) verifiable connection between token frequency and stepsizes in embedding learning problems.

---

### Official Review · Reviewer_xBEu · 2021-10-31

**Correctness:** 3
**Technical Novelty And Significance:** 3
**Empirical Novelty And Significance:** 2
**Recommendation:** 6
**Confidence:** 3

**Main Review:**

Strengths:
- The paper addresses a very interesting problem about incorporating frequency information into the optimizer for embedding learning in recommender systems.
- The paper is well-written and the proposed method is well-motivated.
- The paper demonstrates that the proposed method has provable benefits over SGD.

Weakness: In my opinion, the paper can be improved a lot by re-framing the pitch and enriching the experiments.
- The title/abs/intro of this paper covers the various areas of embedding learning while the experiments are only on recommendation. I would rather suggest the authors frame the pitch in the area of recommender systems instead of in general embedding learning.
- To echo with the above points, it will be very interesting if the authors also show some empirical evidence of the proposed method on other embedding learning tasks, like word2vec of knowledge graph embeddings.
- In Figure 2 (b), it looks like there is a big jump in the red line at the iteration around 1500. Could you explain what was happening there?
- I wonder if you tune the momentum in SGD.
- Do you have figures of correlation on Movielens-1M? like Figure 3(a) and 3(b)?
- In Figure 4 (d) what's the Y-axis?
- I don't quite get why "In Figure 4c, 4d we compare the training NE curve CF-SGD and Adagrad, we can see that CF-SGD shows faster convergence than Adagrad during training". Looking at Figure 4c, I only see the same convergence rate of CF-SGD and Adagrad.
- It would be interesting to see a curve plot of the learning rate for each token v.s. the frequency of the token.

Minor issues (typos, formats):
- workpiece -> wordpiece?

**Summary Of The Paper:**

This paper proposes a frequency-aware learning algorithm for embedding learning in recommender systems. The idea of incoporating frequency information into the learning algorithm is very interesting to me. The proposed method is very simple and easy to implement with provable benefits.


**Summary Of The Review:**

The paper addresses the problem of adaptive learning rates in recommender systems. The authors propose a novel and simple method to incorporate token frequency information into the learning rate schedule. The authors also show provable benefits over SGD. I think the paper can be further improved by changing the pitch to focus more on recommender systems and enriching the experiments as said in the main reviews.

---

> ### Author Response · Authors · 2021-11-11
> **Response to Reviewer xBEu**
>
> Thank you for the valuable feedbacks on our paper. Here we provide our responses to your remarks and hope these could address your current concerns.
>
> **Complementary experiments on other embedding learning problems:** We gently reiterate that our theoretical developments are our core contributions, which are also completely new in the literature. The results we obtain hold beyond the scope of learning recommendation systems, and the *provable advantage of proposed FA/CF-SGD holds for any embedding learning task as long as the token distribution is imbalanced*. We completely agree that showing additional empirical evidence on other embedding learning problems (e.g. NLP) would be a great supplement for the empirical part of the paper, which we will include in our next version. Meanwhile, we also gently remark that the *current experiments are intended as empirical validation of our developed theories instead of our core contribution*. The core message we aim to deliver through this paper is not to develop an empirically better method for learning recommendation systems, but a new perspective on *efficient algorithmic design that draws (theoretically) verifiable connection between token frequency and stepsizes in embedding learning problems*.
>
>
> **Details on experiments:**
>
> * **Interpretation of Figure 4d:** the Y-axis is the difference of eval NE (measured in \%, the lower the better), when comparing CF-SGD to Adagrad. The NE (normalized entropy) is a popular metric used for industrial recommendation models for its ability to better calibrate the model prediction based on empirical CTR [1]. Please kindly refer to the current draft for the included label for the Y-axis of Figure 4d.
> * **Interpretation of Figure 4c:** Figure 4c shows the training NE curve of CF-SGD and Adagrad. The reason that the training curves of CF-SGD and Adagrad are visually similar in Figure 4c is due to the fact that CF-SGD improves over Adagrad by 0.03-0.04\%, as demonstrated by Figure 4d. We gently highlight that 0.03-0.04\%  *is in fact a very significant improvement on the production model, and can lead to large engagement gains*. This is based on the fact that the both the model, and the hyperparameters of Adagrad algorithm have been tuned extensively in production usage as the workhorse. In fact, any algorithm that has over 0.02\% percent improvement is considered to be significant in production usage.
> * **Plot of learning rate v.s. frequency:** The reason we do not explicitly plot such a relationship is due to the definition of CF-SGD’s learning rate schedule (12). Note (12) already specifies the exact relationship between learning rate and token frequency is specified in equation (12). We do not see the clear need of visualization given the exact correspondence specified in (12).
> * **Correlation plot:** The correlation of Movielens-1M dataset is presented in Figure 3a, 3b. We will also include the correlation of Criteo dataset in the updated appendix.
> * **Momentum of SGD:** We gently remark that we do not consider momentum SGD. That is, the SGD in the experiments is the standard constant stepsize SGD without momentum. In fact, we compare against Adagrad which is well known to be the state of the art for training industrial recommendation models and outperforms momentum SGD significantly. In addition,  momentum SGD does not necessarily achieve better performance over SGD in various benchmark machine learning tasks, kindly refer to [2].
> Finally, CF-SGD in comparison also does not have any momentum term in the update.
> * **Validation AUC for Movielens-1M:** The jump in the validation AUC in Figure 2b is not due to any artificial/sudden changes in the CF-SGD. We believe it is mostly related to the dataset, as it does not appear in the Criteo dataset.
>
> **References:**
>
> [1] He, Xinran, et al. "Practical lessons from predicting clicks on ads at facebook." Proceedings of the Eighth International Workshop on Data Mining for Online Advertising. 2014.
>
> [2] Wilson, Ashia C., et al. "The marginal value of adaptive gradient methods in machine learning." arXiv preprint arXiv:1705.08292 (2017).

---

> > ### Comment · Reviewer_xBEu · 2021-11-15
> > **Thank you for the detailed reply!**
> >
> > Thank you for your detailed reply to my questions and efforts in revising the paper.
> >
> > I like the empirical analysis and results presented in the paper. I think they would be a very meaningful contribution to the field of recommender systems. I understand that it is very hard to get 0.03-0.04% in recommender systems and the results presented in the paper is quite satisfactory.
> >
> > However, I do feel that the paper can be well-positioned and more impactful if we can change the narrative of the paper to focus on recommender systems instead of broad embedding learning. Embedding learning in other domains can be quite different in terms of evaluation protocols (0.03-0.04% in knowledge graph embedding learning might be just due to random noise rather than meaning improvement), and optimization methods (Adagrad doesn't necessarily outperform SGD+momemtum, ignoring SGD+momentum would be unfair). Unless we have some solid results on other embedding learning tasks, I feel a bit over-claim if we continue with the current narrative.

---

> > > ### Author Response · Authors · 2021-11-23
> > > **Paper Revised, with additional Word2Vec experiments**
> > >
> > > Thank you for your timely response during our previous discussion. We have updated our paper, with a focus on additional highlights of memory efficiency, improved presentation (figure clarity), and additional experiments on learning Word2Vec embeddings. All changes are marked in blue in the updated draft. Here we briefly list our key changes below for quick reference.
> > >
> > > **Highlight on Memory Efficiency**: We have included additional discussions on the memory efficiency of the proposed FA/CF-SGD compared to other adaptive algorithms including Adam, Adagrad. Our key message to be delivered here is that, unlike Adam/Adagrad which requires at least twice the memory for storing historical information (first/second-order gradient moment), FA/CF-SGD only needs additional memory for storing the frequency. When the embedding tables already consume a significant memory footprint, FA/CF-SGD can be tremendously more memory efficient compared Adam/Adagrad. Please kindly refer to our detailed illustration in the updated experiment section.
> > >
> > > **Word2Vec Experiments**: We have also conducted experiments on the Word2Vec embedding learning task, both the CBOW and Skip-Gram models are considered. Due to space constraints, we have included the detailed experiment setup and results in Appendix B. Our empirical results show that CF-SGD still performs comparably to Adam, while SGD converges significantly slower. We view our updated experiments as strong evidence demonstrating the broader applicability of FA/CF-SGD for embedding learning problems.
> > >
> > > **Figure Clarity**: Based on your suggestions, we have improved figure presentation, especially on the industrial recommendation system (Figure 4c, 4d). Specifically, Figure 4c includes a zoom-in illustration of the training NE curves for CF-SGD and Adagrad, which we hope can make the improvement of CF-SGD more distinct. Figure 4d includes an updated Y-label for the description of the NE difference.
> > >
> > > We sincerely hope our improved presentation, additional highlights,  and updated experiments on word embedding learning can help address your remaining concerns of our draft.

---

> > > > ### Comment · Reviewer_xBEu · 2021-11-23
> > > > **Thanks for the additional word2vec experiment!**
> > > >
> > > > Thank you for the additional word2vec experiments! I have raised scores given your dedicated efforts in adding word2vec experiments, and the memory efficiency pitch. I really appreciate your efforts in enhancing the experiments though the benefit of CF-SGD over adagrad/adam is still not quite clear (in the updated word2vec experiment, CF-SGD runs comparably as Adam in Skip-Gram setting but poorly in CBOW setting). However, the pitch on memory efficiency does sound appealing. I would suggest providing some empirical statistics on memory footprint in the experimental section and maybe change the narrative by emphasising memory footprint instead of benefits in general embedding learning.

---

> > > > > ### Author Response · Authors · 2021-11-23
> > > > > **Thank you for the encouraging comments!**
> > > > >
> > > > > We really appreciate your encouraging comments, on acknowledging our efforts in providing supplementary experiments, and providing additional constructive suggestions on our paper presentation. Further changes in the context based on your suggestions will be included in our future draft, we will also include a more comprehensive empirical comparison on NLP tasks in the future draft.

---

### Official Review · Reviewer_42Qf · 2021-11-01

**Correctness:** 4
**Technical Novelty And Significance:** 3
**Empirical Novelty And Significance:** 3
**Recommendation:** 6
**Confidence:** 2

**Main Review:**

Minor concerns:

1. Why are the lines in Figure 4c identical?
2. It might be better to zoom in on the y-axis for training loss and AUCs in a smaller region. It is hard to distinguish each method in the current y-axis scale.
3. How robust is each method towards random seeds and hyper-parameters? Having a variance and confidence interval might help.
4. Any results for languages?


**Summary Of The Paper:**

This paper proposes a counter-based learning-rate scheduler for SGD. This algorithm is designed based on the long tail distribution in recommendations and languages. The proposed algorithm enjoys a theoretic guarantee unlike other adaptive learning rate methods, e.g., Adams. Simultaneously, the authors demonstrated that the proposed algorithm achieved comparable empirical performance in two recommendation datasets. Overview, the paper is well-written, which has presented the contributions clearly.


**Summary Of The Review:**

This paper is well-written and made solid contributions for adaptive learning-rate in long-tail distribution.

---

> ### Author Response · Authors · 2021-11-11
> **Response to Reviewer 42Qf**
>
> We greatly appreciate your positive remarks on our developed results. Here we provide our responses to your detailed comments.
>
> **Regarding Figure 4c:**
> Figure 4c shows the training NE curve of CF-SGD and Adagrad. The reason that the training curves of CF-SGD and Adagrad are visually similar in Figure 4c is due to the fact that CF-SGD improves over Adagrad by 0.03-0.04\%, as demonstrated by Figure 4d. We would like to further highlight that 0.03-0.04\% *is in fact a very significant improvement on the production model, and can lead to large engagement gains*. This is based on the fact that both the model and the hyperparameters of Adagrad algorithm have been tuned extensively in production usage as the workhorse. In fact, any algorithm that has over 0.02\% percent improvement is considered to be statistically significant.
>
> **Clarity of figures:** Thanks for pointing out the clarity issue of the figures, we will create new plots with better visualization in our next version.
>
> **Random seeds and hyper-parameters:** Thanks for the valuable suggestion, we will conduct repeated runs and report mean-variance training curves in our updated version.  We would like to gently remark that in our experiments, we do not vary the random seed and cherry-pick our results. All the experiments use the same global random seed. In addition, the hyperparameter of CF-SGD is not extensively tuned especially on the ultra-large production model, due to the limit of training resources. Yet already the CF-SGD shows benefits over the carefully tuned Adagrad in production usage.
>
> **Results for language models:** Thanks for pointing out the language model, we believe this would also be a great supplement to our experiment section and will include in our revised version. Meanwhile, as we have pointed out in our response to Reviewer xBEu, our theoretical developments are our core contributions, which are also completely new in the literature. The results we obtain hold beyond the scope of learning recommendation systems, and the provable advantage of proposed FA/CF-SGD holds for any embedding learning task as long as the token distribution is imbalanced. The current experiments are intended as empirical validation of our developed theories instead of our core contribution. The core message we aim to deliver through this paper is a new perspective on efficient algorithmic design that draws (theoretically) verifiable connections between token frequency and stepsize in embedding learning problems.

---

### Official Review · Reviewer_8cB1 · 2021-11-03

**Correctness:** 3
**Technical Novelty And Significance:** 3
**Empirical Novelty And Significance:** 3
**Recommendation:** 8
**Confidence:** 3

**Main Review:**

The paper investigates the idea of incorporating frequency into SGD for embedding learning, which makes a lof of sense and hasn't been explored before to my knowledge. The theoretical results is interesting in that the proposed method outperforms SGD *WHEN* skewed distribution appeared. And the good thing is that for frequent users/items, the convergence rate remains the same, but much improved for infrequent ones. The empirical results are also very interesting: (i) reveals that Adam/Adagrad mostly implicitly capture the frequency information in momentum; (ii) show great performance improvement over SGD, and very competitive performance against adaptive methods.

I just have a few comments/questions:

(i) it'd be great to have a slice performance analysis, which may reveal that the improvement in infrequent items is even larger.
(ii) for infrequent items, whether a high convergence rate will contribute to overfitting? though the generalization property is beyond the scope of the paper.
(iii) it seems the hidden width of DeepFM is `(16, 16) `, which seems too small to me? Will the analysis/results change for large NNs?




**Summary Of The Paper:**

The paper proposes a frequency-based SGD method, where the learning rate is inversely proportional to the token frequency. Theoretical results show the proposed methods outperform SGD in skewed distributions, and empirically verify the effectiveness of the proposed method, and reveal the similarity between Adam/AdaGrad's momentum and frequency.



**Summary Of The Review:**

The paper nicely introduces frequency information into SGD, with various interesting theoretical analysis and empirical results.

---

> ### Author Response · Authors · 2021-11-11
> **Response to Reviewer 8cB1**
>
> We greatly appreciate your kind remarks on our theoretical developments and empirical results. Here we provide our responses to your detailed comments/questions:
>
> **Slice performance analysis**: This is indeed a great suggestion. We will include in our updated appendix the slice performance analysis for the academic datasets.
>
> **Overfitting**: As we have shown in Figures 2,3,4, we see faster convergence for both training loss and validation AUC for academic datasets. All the algorithms reach the same final AUC performance, suggesting there is no overfitting created by the proposed CF-SGD. More importantly, for the industrial recommendation system (multiple terabytes of model size and 25 billion training examples), we not only observe faster training performance (measured in training NE, see Figure 4c, 4d), but also obtain improved evaluation performance on the fresh 2.5 billion (1 day)  examples. We gently remark that for the industrial recommendation model, an improvement above 0.02\% is considered to be significant given numerous historical iterations/improvements of the model, which can lead to large engagement gains.
>
>
> **Hidden width:** The choice of hidden width 16 in DeepFM is quite standard, and is also the standard choice in the popular recommendation model codebase Pytorch-FM [1]. Our analysis and our results hold for arbitrary network structure. We will include the additional empirical results for hidden network widths if required.
>
>
> **References:**
> [1] https://github.com/rixwew/pytorch-fm

---

### Author Response · Authors · 2021-11-23
**Paper revised, Word2Vec experiments included**

We deeply appreciate each reviewer for your constructive feedback on our current draft.  We have made updates to our draft accordingly, with a focus on additional highlights of memory efficiency, improved presentation (figure clarity), and additional experiments on learning word embeddings. All changes are marked in blue in the updated draft. Here we briefly list our key changes below for quick reference.

**Highlight on Memory Efficiency**: We have included additional discussions on the memory efficiency of the proposed FA/CF-SGD compared to other adaptive algorithms including Adam, Adagrad. Our key message to be delivered here is that, unlike Adam/Adagrad which requires at least twice the memory for storing historical information (first/second-order gradient moment), FA/CF-SGD only needs additional memory for storing the frequency. When the embedding tables already consume a significant memory footprint, FA/CF-SGD can be tremendously more memory efficient compared Adam/Adagrad. Please kindly refer to our detailed illustration in the updated experiment section.

**Word2Vec Experiments**: We have also conducted experiments on the Word2Vec embedding learning task, both the CBOW and Skip-Gram models are considered. Due to space constraints, we have included the detailed experiment setup and results in Appendix B. Our empirical results show that CF-SGD still performs comparably to Adam, while SGD converges significantly slower. We view our updated experiments as strong evidence demonstrating the broader applicability of FA/CF-SGD for embedding learning problems.

**Figure Clarity**: We have improved figure presentation, especially on the industrial recommendation system (Figure 4c, 4d). Specifically, Figure 4c includes a zoom-in illustration of the training NE curves for CF-SGD and Adagrad, which we hope can make the improvement of CF-SGD more distinct. Figure 4d includes an updated Y-label for the description of the NE difference.

---

### Decision · Program_Chairs · 2022-01-20

**Decision:**

Accept (Poster)

**Comment:**

The paper provides a new learning technique for problems that require learning embeddings. In particular, the authors analyze a technique that takes into account the frequency of items in an embedding layer to modify the learning rate for each embedding. The paper provides a theoretical analysis of this approached and contrasts it to that of SGD. It also provides experiments validating this approach empirically.

The reviewers agree that the paper provides a simple yet effective method, based on realistic assumptions (non-uniform frequencies). In addition, the paper seems to be well written and easy to follow.
One issue raised in the reviews was about the focus of the paper, and the fact that the experiments are limited to recommendation systems even though the method is claimed to be generic for any model requiring embeddings. During the rebuttal the authors provided experiments for an NLP task that show favorable results to the new technique in another regime. Given the overall positive feedback and this new evidence validating the proposed method, I recommend accepting the paper.